# Nonlinear effects of the stratospheric Quasi-Biennial Oscillation on ENSO modulating PM<sub>2.5</sub> over the North China Plain in early winter

Xiadong An<sup>1,2,3</sup>, Wen Chen<sup>1,2</sup>, Tianjiao Ma<sup>1,2\*</sup>, Lifang Sheng<sup>3\*</sup>

- <sup>1</sup>Yunnan International Joint Laboratory of Monsoon and Extreme Climate Disasters/Yunnan Key Laboratory of Meteorological Disasters and Climate Resources in the Greater Mekong Subregion, Yunnan University, Kunming, 650500, China
  - <sup>2</sup> Department of Atmospheric Sciences, Yunnan University, Kunming, 650500, China
  - <sup>3</sup>Department of Marine Meteorology, College of Oceanic and Atmospheric Sciences, Ocean University of China, Qingdao, 266100, China
- 10 Correspondence to: Tianjiao Ma (matianjiao@ynu.edu.cn), Lifang Sheng (shenglf@ouc.edu.cn)

Abstract. The North China Plain (NCP) experiences severe air pollution, with PM<sub>2.5</sub> (fine particulate matter with an aerodynamic diameter ≤ 2.5 µm) as the primary pollutant, especially in early winter (November to December). The PM<sub>2.5</sub> concentrations in this period are significantly modulated by the El Niño-Southern Oscillation (ENSO). In this study, we have found that the stratospheric Quasi-Biennial Oscillation (QBO) exerts a nonlinear impact on the relationship between ENSO and PM<sub>2.5</sub> concentrations over the NCP in early winter. During the easterly QBO (EQBO) phase, ENSO's influence on PM<sub>2.5</sub> concentration is stronger compared to the westerly QBO (WQBO) phase. In El Niño and EQBO years, PM<sub>2.5</sub> concentrations rise due to meteorological factors like a shallower boundary layer, higher relative humidity, and intensified southerly wind anomalies. Conversely, during La Niña and EQBO years, PM<sub>2.5</sub> levels decrease due to opposite meteorological conditions. The study attributes these changes to planetary wave dynamics. During El Niño and EQBO years, upward-propagating planetary waves in mid-latitudes enhance upper-level divergence over Eurasia, strengthening westerlies. These westerlies guide Rossby wave trains into Northeast Asia, forming a strong anomalous anticyclone that worsens air pollution over the NCP. In La Niña and EQBO years, downward-propagating planetary waves induce divergence in sub-polar regions, strengthening westerlies that facilitate La Niña-related wave trains. These wave trains trigger cyclonic circulation over Northeast Asia, improving air quality in the NCP. These findings underscore the complex interplay between ENSO, QBO, and atmospheric dynamics in shaping regional air pollution.

## 1 Introduction

15

Air pollution dominated by  $PM_{2.5}$  (fine particulate matter with an aerodynamic diameter  $\leq 2.5 \,\mu m$ ) poses enormous risks to human health and socioeconomic development (e.g., Shi et al., 2021; Geng et al., 2021). The North China Plain (NCP), one of the most densely populated regions in China, has been severely troubled by  $PM_{2.5}$  pollution, particularly in winter months. The declining trend of  $PM_{2.5}$  concentrations in recent years appears to be slowing, despite the Chinese government's comprehensive emission control measures implemented since the 2010s (Silver et al., 2025). Previous studies have

highlighted that wintertime PM<sub>2.5</sub> pollution in eastern China is primarily driven by both emissions and meteorological conditions (Dang and Liao, 2019; Wu et al., 2022; Jing et al., 2023; Dai et al., 2023). Notably, the interannual variability of PM<sub>2.5</sub> concentrations over the NCP is largely influenced by meteorological conditions (Dang and Liao, 2019), particularly large-scale atmospheric circulation patterns and associated climate factors (An et al., 2020, 2022a, 2022b, 2023a; Wang et al., 2020; Yin et al., 2021).

The El Niño-Southern Oscillation (ENSO), the most prominent interannual variability in the atmosphere-ocean interactions globally (Bjerknes, 1969; Rasmusson and Carpenter, 1982; Philander et al., 1984), has a profound impact on global weather and climate (Alexander, 1992; Zhang et al., 1996; Wang et al., 2000; Zhang and Akimasa, 2002; Yuan, 2004; Zhou et al., 2014). Given its crucial role in shaping global weather and climate, extensive research has been conducted to explore whether ENSO influences air pollution in the NCP. For example, some studies suggested a significant relationship between ENSO and air pollution over the NCP (e.g., Chang et al., 2016; Jeong et al., 2018; Sun et al., 2018; Zhang et al., 2019; Xie et al., 2021; Zeng et al., 2021), while others argued there is little to no correlation (e.g., Li et al., 2017; Zhao et al., 2018a, 2018b; Cheng et al., 2019; He et al., 2019). Recently, building on a synthesis of these conflicting findings, research has increasingly suggested that ENSO does indeed affect air pollution over the NCP, particularly during the November to January period (Zhao et al., 2022; An et al., 2022a, 2022b, 2023a). The physical mechanisms behind this influence are now better understood. One explanation is that ENSO impacts air pollution in the Beijing-Tianjin-Hebei region (Fig. S1) by modulating the Hadley circulation during November-December (Zhao et al., 2022). Another mechanism involves ENSOinduced wave trains along the mid- and high-latitudes pathways in the Northern Hemisphere, as well as the Indo-Pacific great circle route, which trigger a Northeast Asian anomalous anticyclone, thereby influencing air pollution over the NCP from November to January (An et al., 2022a, 2022b, 2023a). This new body of evidence helps the debate regarding the role of ENSO in air pollution over the NCP.

In addition to ENSO, the stratospheric Quasi-Biennial Oscillation (QBO) is another crucial climatic factor influencing global atmospheric circulation (Holton and Tan, 1980; Chen and Li, 2007; Wei et al., 2007; Rao et al., 2020; Cai et al., 2022; Ern et al., 2023; Kang et al., 2023; Zhang and Zhou, 2023) and regional air quality (Liang and Tao, 2017; Lu et al., 2022; Li et al., 2023; Zhang et al., 2025). Both the QBO and ENSO originate in the tropics, with the former being a stratospheric phenomenon and the latter a tropospheric one. Notably, the QBO and ENSO may interact with each other in a linear manner (Wang et al., 2023), and together, they influence atmospheric circulations over the North Atlantic in winter (Ma et al., 2023), and winter climate over Eurasia (Chen et al., 2005, 2007; Ma et al., 2021; Zhang et al., 2023). For example, Ma et al. (2021) reported that the easterly QBO (EQBO) weakens East Asian winter monsoon during November–December. The weakening of the monsoon tends to deteriorate air quality in the NCP (An et al., 2020; Zhao et al., 2021). This raises the question of whether the QBO modulates the relationship between ENSO and air pollution over the NCP during early winter.

55

In this study, we examine the evidence that the QBO nonlinearly modulates the relationship between ENSO and PM<sub>2.5</sub> concentrations over the NCP during early winter (November–December, hereafter referred as ND). We analyse the

characteristics of PM<sub>2.5</sub> concentrations and the associated meteorological conditions during different ENSO and QBO composites and explore the underlying physical mechanisms. The remainder of this paper is organized as follows: Section 2 describes the datasets and methods; Section 3 discusses nonlinear effects of the QBO on the connection between ENSO and PM<sub>2.5</sub> concentration; The underlying physical mechanisms are explored in section 4; Finally, a brief summary and discussion of the results are provided in section 5.

## 2 Data and methods

## 2.1 Data

70

The monthly gridded near-surface  $PM_{2.5}$  dataset provided by Yang (2020) has a horizontal resolution of  $1^{\circ}\times1^{\circ}$  and covers the period from 1980 to 2019. This dataset was reconstructed by Li et al. (2021) using a spatiotemporal random forest model based on atmospheric visibility observations and other auxiliary data. According to Li et al. (2021), the monthly  $PM_{2.5}$  concentrations show excellent agreement with ground-based measurements, with a coefficient of determination of 0.95 and a mean relative error of 12%. This dataset has been widely used in studies of air pollution in China (e.g., An et al., 2022b, 2022c; Zhang et al., 2023, 2024). To further validate the results based on  $PM_{2.5}$  data provided by Yang (2020), we also used monthly  $PM_{2.5}$  data spanning 1960 to 2020, as provided by Zhong et al. (2022a, 2022b). For further details on the dataset provided by Zhong et al. (2022a) please refer to Zhong et al. (2022b). To minimize the influence of emissions on the interannual variability of air pollution, we removed the least-squares linear trend from the original  $PM_{2.5}$  data in this study. After removing the quadratic trend, the correlation coefficient between the observed  $PM_{2.5}$  concentrations and the  $PM_{2.5}$  concentrations from emissions is 0.08 (p-value = 0.69), whereas for the original observed  $PM_{2.5}$  concentrations, the correlation coefficient with emissions is 0.40 (p-value = 0.03). This indicates that removing the quadratic trend partially eliminates the influence of emissions on the observed  $PM_{2.5}$  concentrations.

The monthly atmospheric reanalysis products, including geopotential height, zonal and meridional winds, and relative humidity, were obtained from the ERA5, the latest generation of the European Centre for Medium-Range Weather Forecasts (ECMWF) (Hersbach et al., 2020). These reanalysis products have a horizontal resolution of  $0.25^{\circ} \times 0.25^{\circ}$  and a vertical resolution of 37 levels from 1000 hPa to 1 hPa, starting from 1940. In addition, the reanalysis data from the NCEP–DOE AMIP-II reanalysis (Kanamitsu et al., 2002), provided by the National Centers for Environmental Prediction (NCEP)/National Center for Atmospheric Research (NCAR), are used to further validate the robustness of the results. The NCEP/NCAR atmospheric reanalysis dataset features a horizontal resolution of  $2.5^{\circ} \times 2.5^{\circ}$ , 17 vertical levels from 1000 hPa to 10 hPa, and spans the period from 1979 to the present. Monthly boundary layer height data were downloaded from the ERA5, with a horizontal resolution of  $0.25^{\circ} \times 0.25^{\circ}$  since 1940. The ENSO index was obtained from Climate Prediction Center (CPC) of the National Weather Service (NWS) in the United States and the QBO index was provided by Freie Universität Berlin (FUB). The sea surface temperature (SST) data used in this study are from the National Oceanic and

Atmospheric Administration (NOAA) Extended Reconstructed SST (ERSST) version 5 dataset, which provides global monthly SST at a spatial resolution of  $2.0^{\circ} \times 2.0^{\circ}$  from January 1854 to the present (Huang et al., 2017).

# 2.2 Methods

100

105

To capture the characteristics of the QBO, this study defines the 10-hPa zonal wind speed as the QBO index. The mean wind speeds for November–December greater than 2 m s<sup>-1</sup> and less than –2 m s<sup>-1</sup> are categorized as the westerly QBO phase (WQBO) and EQBO, respectively. The classification of cold and warm phases of ENSO events is based on the National Climate Centre of China website (http://cmdp.ncc-cma.net/cn/index.htm) and the website provided by Jan Null (see https://ggweather.com/enso/oni.htm and Text S1 for more details). When selecting ENSO events, the focus is on whether the periods from October to December and November to January correspond to El Niño or La Niña. Given that both ENSO and QBO exert their strongest climatic impacts on East Asia during November and December (ND), this study particularly emphasizes the season of ND, as highlighted in previous studies (Ma et al., 2021; Zhao et al., 2022). Figure 1 illustrates the spatial and temporal evolution of the QBO and ENSO during the research period of this paper. The different combinations of ENSO and the OBO are summarized in Table 1.

In this study, we apply the bootstrap method (Austin and Tu, 2004) to obtain more samples. Specifically, we randomly resample from the selected data, allowing repeated selection of any sample. This process is repeated 1,000 times to compute the mean of the resampled samples.

To examine the propagation of tropospheric Rossby wave trains responsible for air pollution over the NCP, the horizontal wave activity flux is calculated following the method outlined by Takaya and Nakamura (2001):

$$\boldsymbol{W} = \frac{p cos \varphi}{2|\boldsymbol{U}|} \cdot \begin{pmatrix} \frac{U}{a^2 cos^2 \varphi} \left[ \left( \frac{\partial \psi'}{\partial \lambda} \right)^2 - \psi' \frac{\partial^2 \psi'}{\partial \lambda^2} \right] + \frac{V}{a^2 cos \varphi} \left[ \frac{\partial \psi'}{\partial \lambda} \frac{\partial \psi'}{\partial \varphi} - \psi' \frac{\partial^2 \psi'}{\partial \lambda \partial \varphi} \right] \\ \frac{U}{a^2 cos \varphi} \left[ \frac{\partial \psi'}{\partial \lambda} \frac{\partial \psi'}{\partial \varphi} - \psi' \frac{\partial^2 \psi'}{\partial \lambda \partial \varphi} \right] + \frac{V}{a^2} \left[ \left( \frac{\partial \psi'}{\partial \varphi} \right)^2 - \psi' \frac{\partial^2 \psi'}{\partial \varphi^2} \right] \end{pmatrix}, \tag{1}$$

where  $\boldsymbol{W}$  is the wave activity flux with unit of m<sup>2</sup> s<sup>-2</sup>;  $\psi$  (=  $\Phi/f$ ) is the geostrophic stream function, in which  $\Phi$  (m) is geopotential height, f (=  $2\Omega\sin\phi$ ) is the Coriolis parameter;  $\lambda$  is the longitude;  $\varphi$  is the latitude;  $\alpha$  is the radius of the earth; p is the normalized pressure (pressure per 1000 hPa); U (= (U,V); m s<sup>-1</sup>) is the basic flow. The primes denote anomalies after the removal of the climatological mean for the period 1979–2020.

To diagnose the interaction between the troposphere and stratosphere, we computed the Eliassen-Palm (EP) flux (Edmon et al., 1980; Andrews et al., 1987). The quasi-geostrophic EP flux in spherical geometry was calculated following the method outlined by Chen et al. (2003):

$$F_{v} = -\rho a cos \varphi \overline{u'v'}, \tag{2}$$

$$F_z = \rho a cos \varphi \frac{Rf}{HN^2} \overline{v'T'}, \tag{3}$$

$$D_F = \frac{\nabla \cdot \vec{F}}{\rho a cos \varphi},\tag{4}$$

where  $\vec{F}(F_y, F_z)$  is the components of the EP flux;  $D_F$  is the EP flux divergence;  $\rho$  is the air density;  $\alpha$  is the earth's radius;  $\varphi$  is the latitude; R is the gas constant; f is the Coriolis parameter; H is a constant-scale height (7 km); N is the buoyancy frequency; u and v are the zonal and meridional wind; T is the air temperature; the primes and overbars denote zonal deviations and means. The EP flux is scaled according to Edmon et al. (1980) for pressure coordinates.

Given that  $PM_{2.5}$  concentrations over the NCP are affected by multiple factors, such as ENSO and the QBO, to further highlight the impact of the QBO on  $PM_{2.5}$  concentrations over the NCP during ENSO events, we applied multivariate linear regression analyses following the approach of An et al. (2023a). The regression model is formulated as follows:

$$re - NAAA = 0.36 \times BKS + 0.40 \times Ni\tilde{n}o3 + 0.19 \times QBO. \tag{5}$$

Here, re-NAAA represents the reconstructed northeast Asian anomalous anticyclone (NAAA) index (unit: m), a key circulation pattern influencing PM<sub>2.5</sub> pollution in the NCP, which was identified by An et al (2023a). BKS, Niño3, and QBO denote the Barents-Kara Sea sea-ice index, the ENSO index, and the QBO index, respectively. Among them, ENSO and Arctic sea ice have already been identified by An et al. (2023a) as key factors influencing the NAAA.

## 3 Nonlinear effects of the OBO on the relationship between ENSO and PM<sub>2.5</sub> concentrations

Figure 2 shows the distributions of PM<sub>2.5</sub> concentration anomalies in eastern China during different combinations of ENSO and QBO events. It is evident that during La Niña and the EQBO phase, there is a significant reduction in PM<sub>2.5</sub> concentration over the NCP, while during El Niño and EQBO composites, PM<sub>2.5</sub> concentrations increase substantially (Figs. 2a, 2c and 3). Specifically, during La Niña events combined with the EQBO (WQBO) phase, the regional averaged PM<sub>2.5</sub> concentration over the NCP (32°N–42°N, 110°E–120°E) decreases by −4.10 (+0.35) μg m<sup>-3</sup>, whereas during El Niño events with the EQBO (WQBO) phase, the concentration increases by +3.41 (+0.54) μg m<sup>-3</sup>. These results indicate that the QBO phase significantly modulates the ENSO–PM<sub>2.5</sub> relationship over the NCP, highlighting its non-negligible role in regional air quality variability. In addition, the composited PM<sub>2.5</sub> anomalies shown in Fig. 2b and 2d during the WQBO phase exhibit opposite signs across most parts of China, while in the NCP, which is the main focus of this study, the PM<sub>2.5</sub> anomalies do not change much regardless of whether ENSO is in the La Niña or El Niño phase. These conclusions are also supported by daily observations (Fig. S2). This suggests a potential relationship between the EQBO and PM<sub>2.5</sub> concentrations over the NCP in early winter during ENSO events.

Previous studies have indicated that ENSO events significantly modulate early-winter air pollution (i.e., higher PM<sub>2.5</sub> concentration) over the NCP (Chang et al., 2016; Zhao et al., 2022; An et al., 2022a, 2022b, 2023a). It is noteworthy that in non-ENSO years, regardless of whether the QBO is in easterly or westerly phase, no significant change in PM<sub>2.5</sub> concentration is observed (Fig. 2e and f), indicating that ENSO events are the key factor driving changes in PM<sub>2.5</sub> concentrations over the NCP. As shown in Fig. 2h, the ENSO index exhibits a significant positive correlation with PM<sub>2.5</sub> concentrations over the NCP. Interestingly, when the linear relationship between the QBO and PM<sub>2.5</sub> concentrations is

removed, the relationship between ENSO and PM<sub>2.5</sub> remains almost identical (Fig. 2g and 2f), suggesting that while the QBO does not directly influence PM<sub>2.5</sub> concentrations, it significantly nonlinearly modulates the relationship between ENSO and PM<sub>2.5</sub> concentrations over the NCP in early winter. It is worth mentioning that the QBO during the periods 2015/2016 and 2019/2020 exhibited distinct evolutions. Removing the QBO data for these periods does not affect our main conclusions (not shown). The results based on PM<sub>2.5</sub> data provided by Zhong et al. (2022) also support these findings (Fig. S3). The question now is whether the observed changes in PM<sub>2.5</sub> concentrations during different combinations of ENSO and EQBO phases can be attributed to concurrent changes in meteorological conditions associated with both ENSO and EQBO during the same period.

To investigate the mechanisms through which meteorological conditions related to ENSO and the EQBO modulate PM<sub>2.5</sub> concentrations over the NCP in early winter, we examine the distributions of 850-hPa wind fields, 925-hPa relative humidity, and atmospheric boundary layer height during ENSO and EQBO composites. The results clearly show that during La Niña and EQBO composites (and similarly, El Niño and EQBO composites), the lower-level anomalous wind fields over the NCP shift distinctly towards northerly (southerly) winds, which correspond to an intensification (weakening) of the East Asian winter monsoon (Fig. 4). This shift in winds enhances (diminishes) atmospheric dispersion conditions for air pollutants, leading to decrease (increase) in PM<sub>2.5</sub> concentrations, as observed over the NCP during early winter in La Niña and EQBO composites (El Niño and EQBO composites). Additionally, an analysis of atmospheric boundary layer height anomalies reveals that during La Niña and EQBO composites (El Niño and EQBO composites), the northern part of the NCP experiences higher (lower) boundary layer heights (Fig. 5a and 5c), which favors (hinders) the vertical dispersion conditions of air pollutants in the region, ultimately promoting the accumulation of PM<sub>2.5</sub>. In contrast, during La Niña and WQBO composites (El Niño and WQBO composites), no significant change in atmospheric boundary layer height is observed over the NCP, resulting in little to no change in PM<sub>2.5</sub> concentrations.

In addition to dynamic factors, atmospheric thermal conditions (i.e., humidity) also contribute to the variation in PM<sub>2.5</sub> concentrations over the NCP during early winter, particularly in La Niña (El Niño) and EQBO composites. As shown in Figs. 5e and 5g, relative humidity over the NCP is lower (higher) during La Niña (El Niño) and EQBO composites compared to La Niña (El Niño) and WQBO composites, despite the absence of precipitation (Fig. S4). Previous studies have shown that higher relative humidity during non-precipitation periods favors the hygroscopic growth of particulate matters (Zhou et al., 2011). An at al. (2020) reported that during episodes of severe air pollution (i.e., higher PM<sub>2.5</sub> concentration) in the NCP, relative humidity in the region is indeed higher. These findings suggest that lower (higher) relative humidity under non-rainy conditions during La Niña (El Niño) and EQBO composites contributes to the occurrence of lower (higher) PM<sub>2.5</sub> concentrations over the NCP in early winter.

In summary, both atmospheric dynamic dispersion and thermal conditions during La Niña (El Niño) and EQBO composites favor lower (higher) PM<sub>2.5</sub> concentrations over the NCP. The remaining questions are which large-scale circulation patterns modulate these meteorological conditions, and how do these relate to ENSO and the QBO.

# 4 Physical mechanisms

210

215

Given the important role of meteorological conditions and their associated large-scale circulations in air pollution, this section mainly elaborates on the influence of meteorological factors (Dang and Liao, 2019; Yin et al., 2021). To elucidate the physical mechanisms underlying the nonlinear modulation of the relationship between early-winter PM<sub>2.5</sub> concentration over the NCP and ENSO by the EQBO, we diagnosed the propagation characteristics and causes of the associated largescale Rossby wave trains. As depicted in Fig. 6a, during La Niña and EQBO composites, northeast Asia exhibits a 195 significant negative geopotential height anomaly at 500 hPa, corresponding to a stronger cyclonic circulation anomaly. Conversely, during El Niño and EQBO composites, northeast Asia displays a notable positive geopotential height anomaly, indicative of a strong anticyclonic circulation anomaly (Fig. 6c). An et al. (2023b) identified the cyclonic circulation anomaly in northeast Asia as a key factor leading to a decrease in PM<sub>2.5</sub> concentrations over the NCP during early winter. Conversely, the anticyclonic circulation anomaly in northeast Asia (referred to as the NAAA by An et al. (2022b)) is a 200 crucial system responsible for the increase in early-winter PM<sub>2.5</sub> concentration over the NCP (Zhong et al., 2019; An et al., 2020, 2022a, 2022b, 2022c, 2023a). These circulation systems primarily influence PM<sub>2.5</sub> concentrations over the NCP by modulating lower-level wind fields, near-surface relative humidity, and boundary layer height. In contrast, during La Niña and WQBO composites (El Niño and WQBO composites), the intensity of the cyclonic circulation anomaly (anticyclonic circulation anomaly) in northeast Asia is weaker, and its position is shifted eastward. These changes are unfavourable for 205 inducing changes in meteorological conditions that affect PM<sub>2.5</sub> concentrations in the NCP.

Delving deeper, we find that during La Niña and EQBO composites, the northeast Asian cyclonic anomaly is driven by a Rossby wave train originating from northeast Pacific and traversing the Eurasian sub-polar region (Fig. 6a). Similarly, the NAAA during El Niño and EQBO composites results from a Rossby wave train passing through the mid- and lower-latitudes of the Eurasian continent, also originating from northeast Pacific (Fig. 6c). Previous studies have indicated that ENSO events can trigger wave trains similar to those illustrated in Figs. 6a and 6c (Yu and Sun, 2020; An et al., 2022a, 2023a). Notably, the position of the Rossby wave train in La Niña and EQBO composites is more northerly compared to during El Niño and EQBO composites, which may be linked the positioning of the waveguides that channel the energy of these wave trains (Hoskins and Ambrizzi, 1993). As a contrast, during La Niña and WQBO composites (El Niño and WQBO composites), the signals associated with ENSO struggle to reach the northeast Asia region, resulting in weaker and more eastward-shifted circulation anomalies (Figs. 6b and 6d). The distribution of wave activity flux and the anomalous streamfunction further corroborate these results (Fig. 7). Specifically, during La Niña and EQBO composites (El Niño and EQBO composites), ENSO-related signals are more likely to propagate into the northeast Asia region, inducing significant circulation anomalies in the region (Figs. 6 and 7), which in turn affect PM<sub>2.5</sub> concentration anomalies over the NCP by modulating local meteorological conditions (Figs. 2–5).

To further understand why the Rossby waves associated with ENSO can propagate into northeast Asia during ENSO and EQBO composites, we examine the zonal wind distribution in the upper troposphere. As shown in Fig. 8, compared to La

Niña and WQBO composites, there is a significant positive zonal wind anomaly at 250 hPa over the high latitudes of the North Atlantic and the Eurasian continent (purple box in Fig. 8a) in La Niña and EQBO composites, corresponding to an acceleration of westerlies in the subpolar region. This is conducive to the downward dispersion of Rossby wave energy in that region. Similarly, compared to La Niña and WQBO composites, El Niño and EQBO composites show a notable positive zonal wind anomaly in the mid- (purple box on the top in Fig. 8c) and lower- latitudes (purple box on the below in Fig. 8c) of the Eurasian continent, indicating a strengthening of westerlies in these regions, which supports the downstream propagation of Rossby waves. Examining the vertical profile of zonal winds in the Northern Hemisphere (Fig. 9a–d) and over the Eurasian continent (Fig. 9e–h) during different ENSO and QBO composites reveals that the anomalous zonal winds extend across the mid- and upper- levels of the troposphere and lower stratosphere in ENSO and EQBO composites (Figs. 9a, 9c, 9e and 9g), implying a potential connection between the troposphere and stratosphere via troposphere-stratosphere interactions. It is worth noting that the focal point of this study, the QBO, represents a classic climatic factor in the stratosphere. The QBO can modulate the propagation of planetary waves between the troposphere and stratosphere (Ma et al., 2021; Koval et al., 2022).

Given that a close connection between the QBO and extratropical circulations (Holton and Tan, 1980; Kinnersley, 1999; Kinnersley and Tung, 1999; Chen et al., 2007), the propagation of planetary waves, as revealed by the EP flux, indicates that in La Niña and EQBO composites, there is a noticeable anomalous propagation of planetary waves in the high latitudes of the Northern Hemisphere, leading to divergence in the upper levels of the troposphere and thus enhancing zonal westerly winds in that region (Figs. 8a, 9e and 10a). During El Niño and EQBO composites, there is anomalous upward propagation of planetary waves in the high latitudes of the Northern Hemisphere, with components in the equatorial region diverging in the mid-latitudes, resulting in a positive zonal wind anomaly (Figs. 8c, 9g and 10c). These accelerated westerly winds facilitate the propagation of Rossby wave energy, enabling the wave trains associated with ENSO to reach northeast Asia and ultimately induce anomalous circulations in that region. These anomalous circulations modulate local meteorological conditions, thereby affecting PM<sub>2.5</sub> concentrations over the NCP. To further ensure the robustness of our conclusions, we recalculated the EP fluxes using daily reanalysis data and found the results to be largely consistent (Fig. S12).

#### 5 Discussions and conclusions

This study, based on PM<sub>2.5</sub> data constructed by machine learning and reanalysis datasets, employs composite analysis methods to investigate the physical mechanisms underlying the modulation role of the relationship between ENSO and PM<sub>2.5</sub> concentrations in early winter over the NCP by the QBO. The results show that the QBO nonlinearly modulates the ENSO-PM<sub>2.5</sub> connection. Specifically, during the EQBO phase, the influence of ENSO on PM<sub>2.5</sub> concentrations over the NCP is more pronounced compared to the WQBO phase. For example, during El Niño and EQBO composites, there is a noticeable increase in early-winter PM<sub>2.5</sub> concentrations over the NCP relative to climatology, while during La Niña and EQBO composites, a decrease is observed. These changes in PM<sub>2.5</sub> concentrations do not occur in ENSO and WQBO composites.

Further results show that these changes in PM<sub>2.5</sub> concentrations are driven by meteorological conditions associated with ENSO and EQBO, including a shallower boundary layer, higher relative humidity, and stronger southerly wind anomalies during El Niño and EQBO composites (opposite conditions occur during La Niña and EQBO composites).

Through the diagnosis of Rossby waves and planetary waves based on wave activity flux and EP flux, it is found that during El Niño and EQBO years, upward-propagating planetary waves in the mid-latitudes of the Northern Hemisphere induce upper-level divergence over subtropical Eurasia, favouring the strengthening of westerlies. Serving as a waveguide, these westerlies facilitate the propagation of El Niño-related Rossby wave trains into northeast Asia, leading to a strong northeast Asian anomalous anticyclone in the region. This anticyclonic anomaly elevates PM<sub>2.5</sub> concentrations over the NCP by changing the aforementioned meteorological conditions. Conversely, during La Niña and EQBO years, downward-propagating planetary waves induce mid- and upper-level divergence in the sub-polar regions of the Northern Hemisphere, enhancing westerlies in that area. Acting as a waveguide, these westerlies support propagation of La Niña-related wave trains into northeast Asia, triggering cyclonic circulation within the region. This cyclonic circulation leads to a decrease in PM<sub>2.5</sub> concentrations over the NCP by altering meteorological conditions mentioned as previously described. In contrast, WQBO does not support the propagation of the aforementioned planetary waves, and thus does not provide convenience for the propagation of ENSO-related Rossby waves in the troposphere. Consequently, there are no significant atmospheric circulation anomalies in Northeast Asia, and that the variation in PM<sub>2.5</sub> concentrations over the NCP is less pronounced. The analysis based on NCEP data also supports our conclusion (Figs. S5–11).

Most notably, An et al. (2023a) found that El Niño and Arctic sea-ice increases have a synergistic effect on the Northeast Asian anomalous anticyclone, which in turn contributes to PM<sub>2.5</sub> pollution in early winter over the NCP. They developed a simple linear model that can skilfully predict the Northeast Asian anomalous anticyclone up to one month in advance. However, when considering the QBO, the newly developed prediction model exhibits enhanced performance (Fig. S13). For instance, the correlation coefficient between the new linear model and the raw index is 0.64 (significant at 0.05 level with an effective degree of freedom of 14), surpassing the 0.61 correlation coefficient (significant at 0.05 level with an effective degree of freedom of 14) obtained when the QBO is not considered. Compared with our earlier studies (2022a, 2022b, 2023a), this paper further emphasizes the important role of the QBO in influencing the key atmospheric circulation systems responsible for variations of PM<sub>2.5</sub> concentrations over the NCP.

In addition, An et al. (2020, 2022a, 2022b, 2022c) pointed out that when the NCP experiences air pollution, southern China tends to suffer from heavy rainfall, resulting in the phenomenon of "Southern Rainfall-Northern Haze" in eastern China. This southern rainfall is conducive to air pollution over the NCP by triggering an anomalous anticyclone and promoting convective feedback. Interestingly, during El Niño and EQBO composites (La Niña and EQBO composites), rainfall indeed occurs in southern China (Fig. S4). This suggests that the QBO also nonlinearly modulates the connection between ENSO and the "Southern Rainfall-Northern Haze" phenomenon in eastern China.

It is important to note that while this study highlights the nonlinear role of the QBO in modulating the connection between ENSO and PM<sub>2.5</sub> over the NCP based on observational data, the validation using atmospheric models with interactive chemistry are not included in the current manuscript, with the help of these models and further analysis we would be capable to quantify the respective contributions of emissions and meteorology. This aspect should be addressed in future research. Another limitation of the current study is the lack of high-resolution data (e.g., daily or hourly data) to further investigate the nonlinear processes identified, which remains an important direction for future research. In addition, Ray et al. (2020) pointed out that the QBO influences global surface trace gases. Therefore, it would be worthwhile for future studies to investigate how PM<sub>2.5</sub> concentrations in regions beyond the NCP—such as India—are affected by the combined effects of ENSO and the QBO. Moreover, Zhao et al. (2016) found that the Pacific Decadal Oscillation (PDO) can influence the number of haze days in eastern China at a decadal time-scale. Therefore, whether the PDO modulates the impacts of ENSO and the QBO on PM<sub>2.5</sub> concentrations over the NCP in early winter is a topic worthy of further investigation.

# Data availability

ERA5 atmospheric reanalysis data at pressure levels are available at https://doi.org/10.24381/cds.6860a573 (Hersbach et al., 2018; last access: 25 June 2025). Monthly boundary layer height data, as a surface variable, were also obtained from ERA5 (https://doi.org/10.24381/cds.f17050d7, Hersbach et al., 2018; last access: 6 September 2024). NCEP reanalysis 2 data were PSL, Boulder, CO. USA, provided the NOAA/OAR/ESRL from their Web https://psl.noaa.gov/data/gridded/data.ncep.reanalysis2.html (Kanamitsu et al., 2002, last access: 6 September 2021). The monthly PM<sub>2.5</sub> concentration data can be downloaded from https://zenodo.org/records/4293239 (Yang, 2020; last access: 9 April 2021). The OBO index is available at https://www.geo.fu-berlin.de/en/met/ag/strat/produkte/qbo/index.html (last access: 11 June 2025). The ENSO index can be downloaded at https://www.cpc.ncep.noaa.gov/data/indices/ (last access: 7 April 2024). The SST data is publicly available at https://psl.noaa.gov/data/gridded/data.noaa.ersst.v5.html (last access: 22 June 2025).

# **Author contributions**

XA, LS and WC designed the research and carried it out. XA and TM downloaded and analyzed the reanalysis data and prepared all the figures. XA prepared the paper with contributions from all co-authors. LS, WC, and TM revised the paper.

## **Competing interests**

The contact author has declared that none of the authors has any competing interests.

# Acknowledgement

This research has been jointly supported by the National Natural Science Foundation of China (grant nos. W2412059, 42475043 and 42275191), the Yunnan Provincial Science and Technology Department (grant nos. 202505AB350001 and 202403AP140009), and the Yunnan Southwest United Graduate School Science and Technology Special Project (grant no. 202302AP370003). We thank the editor and the two anonymous reviewers for their constructive suggestions that have helped improve this study.

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

Table 1: Years of QBO and ENSO events based on the QBO index and ENSO events in November to December during the period of 1979–2019.

|         | EQBO                                     | Neutral | WQBO                                     |
|---------|------------------------------------------|---------|------------------------------------------|
| El Niño | 1986, 1997, 2002, 2004, 2006, 2009, 2015 | /       | 1979, 1982, 1987, 1991, 1994, 2014, 2018 |
| Neutral | 1980, 1981, 1985, 1990, 1993, 2013, 2019 | /       | 1989, 1992, 1996, 2001, 2003, 2012       |
| La Niña | 1983, 1988, 1995, 1999, 2008, 2011, 2017 | 2000    | 1984, 1998, 2005, 2007, 2010, 2016       |

Figure 1: (a) Time-height cross-section of the monthly mean equatorial zonal winds provided by the FUB (unit:  $m \, s^{-1}$ ); (b–c) Time series of the 10-hPa QBO and Niño3 indices, averaged in the early winter months (Nov–Dec), each point represents the Nov–Dec average for a given year; (d) Regression coefficients of SST anomalies (unit:  $^{\circ}$ C) onto the Niño3 index shown in (c). White dots indicate where values are significant at the 99% confidence level.

Figure 2: Composite patterns of anomalous PM<sub>2.5</sub> concentrations (shading; unit:  $\mu g \, m^{-3}$ ) for (a) La Niña & EQBO, (b) La Niña & WQBO, (c) El Niño & EQBO, (d) El Niño & WQBO, (e) nENSO & EQBO, (f) nENSO & WQBO. (g) and (h) Regression coefficients of anomalous PM<sub>2.5</sub> concentrations onto the Niño3 index (QBO signals removed) and Niño3 index (the original data) shown in Fig. 1b–c. White dots (black grids in (a)–(f)) indicate areas of significance at the 80% (90%) confidence level.

Figure 3: The average-mean PM<sub>2.5</sub> concentration anomalies over the NCP (32°N-42°N, 110°E-120°E) for (a) La Niña & EQBO and La Niña & WQBO; (b) El Niño & EQBO and El Niño & WQBO. Boxplots are drawn based on 1,000 resamples using the bootstrap method.

Figure 4: Composite patterns of wind fields at 850 hPa. Red vectors represent climatological winds, and blue arrows represent anomalous winds for (a) La Niña and EQBO, (b) La Niña and WQBO, (c) El Niño and EQBO, and (d) El Niño and WQBO. Shaded areas in (a-b) and (c-d) represent the differences in 850-hPa wind speed between La Niña & EQBO and La Niña & WQBO, and between El Niño & EQBO and El Niño & WQBO, respectively.

Figure 5: Same as Figure 4, but for (a-d) boundary layer height anomalies (unit: m) and (e-f) 925-hPa relative humidity anomalies (unit: %). White dotted (black grid) areas indicate significant values at the 80% (90%) confidence level.

Figure 6: Composite patterns of geopotential height anomaly at 500 hPa (shading and contours; unit: m) for (a) La Niña and EQBO, (b) La Niña and WQBO, (c) El Niño and EQBO, and (d) El Niño and WQBO. White dotted (black grid) areas indicate significant values at the 80% (90%) confidence level. The black box represents the key region of the northeast Asian anomalous anticyclone, as defined by An et al. (2023a). The green lines represent the Yellow River and the Yangtze River.

Figure 7: Same as Figure 6, but for the perturbation streamfunction (shading; unit: $10^6$  m<sup>2</sup> s<sup>-1</sup>) and wave activity flux (vectors; unit: m<sup>2</sup> s<sup>-2</sup>) at 500 hPa.

Figure 8: Same as Figure 6, but for climatological zonal winds (green contours; unit:  $m \ s^{-1}$ ; interval: 5) and zonal wind anomalies (shading; unit:  $m \ s^{-1}$ ) at 300 hPa. White dotted (black grid) areas indicate significant values at the 80% (90%) confidence level. Purple boxes represent the key regions of zonal wind changes.

Figure 9: Climatological mean of the zonal-mean zonal winds (red contours with an interval of 10 m s<sup>-1</sup>) and composite zonal-mean zonal wind anomalies (blue contours with an interval of 2 m s<sup>-1</sup>) in 0°-360°E for (a) La Niña and EQBO, (b) La Niña and WQBO, (c) El Niño and EQBO, and (d) El Niño and WQBO. (e)-(f) Same as (a)-(d), but for zonal winds averaged in 0°-140°E. Solid (dashed) lines represent positive and negative values, respectively. Grey shaded areas indicate significant values of the composite zonal winds at the 90% confidence level.

Figure 10: Cross-sections of the EP flux (vectors; unit:  $m^2$  s<sup>-2</sup>) for waves 1–2 and its divergence (contours, with red (blue) lines represent divergence (convergence); unit: m s<sup>-1</sup> day<sup>-1</sup>) for (a) La Niña and EQBO, (b) La Niña and WQBO, (c) El Niño and EQBO, and (d) El Niño and WQBO. Heavy and light shaded areas indicate significant values at the 95% and 90% confidence levels, respectively.