# Peer review of "Nonlinear effects of the stratospheric Quasi-Biennial Oscillation on ENSO modulating PM2.5 over the North China Plain in early winter"

_EGUsphere, 2025_

## Author Response (AR1)

Dear reviewer #1,

We sincerely appreciate your careful review of our manuscript and your valuable suggestions for improving the paper. We have thoroughly considered all comments and revised the manuscript accordingly. Below are our point-by-point responses. *Italicized text* indicates the reviewers' comments, while the regular text represents our responses. The specific revisions are highlighted in red, and all corresponding changes have been marked in the manuscript in the same manner.

Sincerely,

Xiadong An

On behalf of all authors

**Anonymous Referee #1**

The paper "Nonlinear effects of the stratospheric Quasi-Biennial Oscillation on ENSO modulating PM2.5 over the North China Plain in early winter" by An et al. investigates the combined effect of the QBO and ENSO on the PM2.5 concentration over the North China Plain. Main findings are that PM2.5 is enhanced for El Nino and westward QBO conditions, while PM2.5 is reduced for La Nina and eastward QBO. This effect is explained by variations in wind speed and direction, boundary layer height, and humidity. Overall, the paper is a thorough study, it is well written, and of interest and relevance for the readership of ACP. The paper is therefore recommended for publication in ACP after minor revisions.

**Response:**

My main comments are:

(1) the authors should include the model equations they assume for their multivariate linear regressions **Response:** Thank you for your helpful suggestion. In response, we have added the explicit form of the multivariate linear regression model used in our analysis to the revised manuscript.

**Lines 124–129:** "To further highlight the impact of the QBO on PM2.5 concentrations over the NCP during ENSO events, we applied multivariate linear regression analyses following the approach of An et

al. (2023a). The regression model is formulated as follows:

$$re - NAAA = 0.36 \times BKS + 0.40 \times Ni\tilde{n}o3 + 0.19 \times QBO. \tag{5}$$

Here, re-NAAA represent the key circulation pattern influencing PM2.5 pollution in the NCP. BKS, Niño3, and QBO denote the Barents-Kara Sea sea-ice index, the ENSO index, and the QBO index, respectively."

(2) you should explain a bit why enhanced humidity would lead to enhanced  $PM_{2.5}$ , but not to precipitation that would wash out air pollution

**Response:** Thank you for your insightful comment. We have addressed this issue in detail in our response to your specific Comment (4) below. Please refer to that section for further explanation.

Please find more Specific and Technical comments below.

Specific comments:

(1) Fig.1d: what is the source of the SST data?

**Response:** Thank you for your comment. We apologize for the omission regarding the description of the data. The SST data used in Fig. 1d are derived from the NOAA Extended Reconstructed SST (ERSST) version 5 dataset, which provides global monthly SST at a spatial resolution of  $2.0^{\circ} \times 2.0^{\circ}$ . We have added this information in the data section and Data availability for clarity.

**Lines 90–92:** "The sea surface temperature (SST) data used in this study are from the National Oceanic and Atmospheric Administration (NOAA) Extended Reconstructed SST (ERSST) version 5 dataset, which provides global monthly SST at a spatial resolution of  $2.0^{\circ} \times 2.0^{\circ}$  from January 1854 to the present (Huang et al., 2017)."

**Lines 299–300:** "The SST data is publicly available at https://psl.noaa.gov/data/gridded/data.noaa.ersst.v5.html (last access: 22 June 2025)."

**Lines 368–370:** "Huang, B., Thorne, P. W., Banzon, V. F., Boyer, T., Chepurin, G., Lawrimore, J. H., Menne, M. J., Smith, T. M., Vose, R. S., and Zhang, H.-M.: NOAA Extended Reconstructed Sea Surface Temperature (ERSST), Version 5 [dataset], doi:10.7289/V5T72FNM, 2017 (last access: 22 June 2025)."

(2) l.116 onward: You should state more clearly in the text that multivariate regression analyses are performed. The underlying linear models should be given as additional equations.

**Response:** Thank you for this helpful suggestion. In response, we have clarified in the text that multivariate linear regression analyses were conducted. Additionally, we have included the corresponding regression model equations to explicitly describe the statistical framework used in our study.

**Lines 124–129:** "To further highlight the impact of the QBO on PM2.5 concentrations over the NCP during ENSO events, we applied multivariate linear regression analyses following the approach of An et al. (2023a). The regression model is formulated as follows:

$$re - NAAA = 0.36 \times BKS + 0.40 \times Ni\tilde{n}o3 + 0.19 \times QBO. \tag{5}$$

Here, re-NAAA represent the key circulation pattern influencing PM2.5 pollution in the NCP. BKS, Niño3, and QBO denote the Barents-Kara Sea sea-ice index, the ENSO index, and the QBO index, respectively."

(3) l.120: In the text you should add the information that the regression coefficients in Figs.2b and 2d have opposite sign in most regions of China, while in the North China Plain (NCP), which is the main focus of your work, PM2.5 anomalies do not change much.

**Response:** Thank you for the valuable comment. We have added a sentence in the text to clarify that the PM2.5 concentrations in Fig. 2b and 2d exhibit opposite signs over most regions of China. Moreover, we now explicitly point out that PM2.5 anomalies over the NCP—the main focus of our study—do not show significant changes under the two ENSO phases.

**Lines 138–140:** "In addition, the composited PM2.5 anomalies shown in Fig. 2b and 2d during the WQBO phase exhibit opposite signs across most parts of China, while in the NCP, which is the main focus of this study, the PM2.5 anomalies do not change much regardless of whether ENSO is in the La Niña or El Niño phase."

(4) l.153: The point with the humidity was not completely clear to me! If humidity is high enough, precipitation would form and wash out air pollution. Please clarify whether or not this is a relevant mechanism during conditions of enhanced humidity in November/December over the NCP. This should

also be clarified in the discussion around 1.246.

Response: Thank you for pointing this out. If the relative humidity is abnormally high and accompanied by precipitation, it can indeed lead to the scavenging of atmospheric particles and thus reduce PM2.5 concentrations. However, in this study, the relative humidity was only higher than the climatological average. Over the North China Plain, the relative humidity was around 30–60% (Figure R1), which does not reach the level typically associated with precipitation. As shown in Figure R2c, no significant rainfall occurred over the region. Nevertheless, this relatively elevated humidity can enhance the hygroscopic growth of aerosol particles (Zhou et al., 2011), but cannot wash out PM2.5. To avoid confusion for readers, we have added further clarification in the revised manuscript.

Line 173: "... La Niña (El Niño) and WQBO composites, despite the absence of precipitation (Fig. S3)."

Line 174: "... higher relative humidity during non-precipitation periods favors the hygroscopic growth of particulate matters ..."

**Lines 176–177:** "These findings suggest that lower (higher) relative humidity under non-rainy conditions during La Niña (El Niño) and EQBO composites ..."

Figure R1. Climatology of relative humidity at 925 hPa during November to December from 1979 to 2020. Blue box represents the North China Plain.

Figure R2: Composite patterns of precipitation (shading; unit: mm day-1) for (a) La Niña and EQBO, (b) La Niña and WQBO, (c) El Niño and EQBO, and (d) El Niño and WQBO. White dotted (black grid) areas indicate significant values at the 80% (90%) confidence level.

(5) l.442: In addition to the white dots, there are also stippled areas in Fig.2. Do these areas refer to another different level of significance? I would suppose that the white dots refer to 80%, and the black grid areas to 90% of significance, like in Fig.5. Please explain and correct!

**Response:** Thank you for your careful review. As you correctly supposed, the white dots and stippled areas indicate the same significance levels as shown in Fig. 5. We have now added some explanation in the revised text.

Line 502: "White dots (black grids in (a)–(f)) indicate areas of significance at the 80% (90%) confidence level."

(6) Fig.3: to avoid confusion, please use same notation as in Fig.2 (ENSO- instead of ENSO-1 and QBO- instead of QBO-1)

**Response:** We appreciate your thorough review. The figure has been revised accordingly, as detailed below:

**Figure 3:**

(7) Fig.6, Fig.7: Green lines represent the Yellow River and the Yangtze River? Please add this information.

**Response:** Yes, you are right. We have added some information as you suggested.

Line 519: "The green lines represent the Yellow River and the Yangtze River."

**Technical corrections:**

(1) l.40: Given its the crucial in shaping global weather and climate -> Given its crucial role in shaping global weather and climate

**Response:** Thank you. We have revised this sentence as you suggested.

Line 39: "Given its crucial role in shaping ..."

(2) reference Austin and Tu (2004) is missing in the References list

**Response:** Thank you for your careful review. We are sorry for missing the reference Austin and Tu (2004). We have added it in the revised manuscript.

**Lines 331–332:** "Austin, P. C., and Tu, J. V.: Bootstrap methods for developing predictive models, The American Statistician, 58(2), 131–137, doi: 10.1198/0003130043277, 2004."

(3) l.105: Palmer -> Palm

**Response:** Thanks. We apologize for the mistake in writing. It has been changed in the revised manuscript.

Line 114: "... the Eliassen-Palm (EP) ..."

(4) l.107: reference Chen et al. (2013) is not in the references list, did you mean Chen et al. (2003)?

**Response:** Thank you for your careful checking. We apologize for this typo. Chen et al. (2013) is Chen et al. (2003). We have changed it.

Line 116: "... method outlined by Chen et al. (2003):"

(5) l.117: It is clearly -> It is evident

**Response:** Thanks. It has been revised.

**Line 132:** "It is evident ..."

(6) l.119: whether is La Nina or El Nino, these is no -> whether ENSO is in the La Nina or El Nino phase, there is no

**Response:** Thank you. We have revised it as you suggested.

Line 140: "... whether ENSO is in the La Niña or El Niño phase, ..."

(7) l.130: The now question is -> The question now is

**Response:** Thank you for your careful review. It has been changed.

Lines 153–154: "The question now is whether ..."

(8) l.138: Fig.3 -> Fig.4

**Response:** Thanks. We regret this writing error. We have revised it.

Line 162: "... winter monsoon (Fig. 4)."

(9) l.142: (Fig. 4a and 4c) -> (Fig. 5a and 5c)

**Response:** Thank you for your detailed comments. We are sorry for the error and have corrected it.

Line 166: "... experiences higher (lower) boundary layer heights (Fig. 5a and 5c), ..."

(10) l.148: 4e and  $4g \rightarrow 5e$  and 5g

**Response:** We appreciate your thorough review. We have revised it.

Line 172: "... 5e and 5g, ..."

(11) l.155: remining -> remaining

**Response:** Thank you for your careful review. We have revised it.

**Line 180:** "The remaining questions ..."

(12) 1.199: 9c 9e and 9g) -> 9c, 9e, and 9g)

**Response:** Thank you for your careful checking. The revision has been made as per your suggestion.

Line 225: "... 9c, 9e and 9g), ..."

(13) l.203: OBO -> QBO

**Response:** Thanks. We changed it.

Line 229: "Given that a close connection between the QBO and ..."

(14) l.442: areas of significant -> areas of significance

**Response:** Thank you for your careful checking. We have revised it according to your suggestion.

Line 502: "... dots indicate areas of significance at the 90% confidence level."

(15) Caption of Fig.7: Is interval = 1, or 0.5? Please check!

**Response:** Thank you for your careful review. The contour interval is 0.5. Since the color bar for shading has already been shown, we have removed the description of the contour interval.

Line 521: "Same as Figure 6, but for the perturbation streamfunction (shading; unit: 106 m2 s-1) ..."

**References**

Zhou, Y., Zhang, H., Parikh, H. M., Chen, E. H., Rattanavaraha, W., Rosen, E. P., Wang, W. X., and Kamens, R.: Secondary organic aerosol formation from xylenes and mixtures of toluene and xylenes in an atmospheric urban hydrocarbon mixture: Water and particle seed effects (II), Atmos. Environ., 45, 3882–3890, doi:10.1016/j.atmosenv.2010.12.048, 2011.

Dear reviewer #2,

We sincerely appreciate your careful review of our manuscript and the valuable suggestions for improving our work. We have thoroughly addressed all comments and revised the manuscript accordingly. Below are our point-by-point responses. *Italicized text* represents the reviewers' comments, while the regular text contains our responses. The specific revisions are highlighted in blue, and all corresponding changes have been marked in the manuscript in the same way.

Sincerely,
Xiadong An
On behalf of all authors

**Anonymous Referee #2**

The authors analyze the effects of the QBO on the ENSO influence on North China Plains pollution. The present work would benefit from substantial improvements, namely in the methodology, quality of presentation and level of discussion. My impression is that the ENSO is the leading factor, and the QBO importance is secondary. This needs to be quantified numerically, beyond the simple Fig. 3 analysis.

Response: Thank you for your thoughtful and constructive comments. You are right. We agree that ENSO plays a dominant role in modulating PM2.5 variability over the North China Plain (NCP), while the QBO exerts a secondary but potentially nonlinear influence. Our intention in this study is to highlight how the QBO modulates the ENSO–PM2.5 relationship, rather than to suggest that the QBO is an equally strong driver. To address your suggestion, we have now included additional quantitative analyses to better assess and compare the contributions of QBO. Specifically, during La Niña events combined with the EQBO (WQBO) phase, the regional averaged PM2.5 concentration over the NCP (32°N–42°N, 110°E–120°E) decreases by −4.10 (+0.35) μg m-3, whereas during El Niño events with the EQBO (WQBO) phase, the concentration increases by +3.41 (+0.54) μg m-3. These results indicate that the QBO phase significantly modulates the ENSO–PM2.5 relationship over the NCP, highlighting its non-negligible role in regional air quality variability. These additional results have also been added

to the revised manuscript to strengthen our conclusions.

**Lines 134–138:** "Specifically, during La Niña events combined with the EQBO (WQBO) phase, the regional averaged PM2.5 concentration over the NCP ( $32^{\circ}N-42^{\circ}N$ ,  $110^{\circ}E-120^{\circ}E$ ) decreases by -4.10 (+0.35)  $\mu g$  m-3, whereas during El Niño events with the EQBO (WQBO) phase, the concentration increases by +3.41 (+0.54)  $\mu g$  m-3. These results indicate that the QBO phase significantly modulates the ENSO–PM2.5 relationship over the NCP, highlighting its non-negligible role in regional air quality variability."

The description and selection of data needs improvement (see detailed comments).

The use of a fairly dated reanalysis product is not justified; while it may be useful to retain it as a reference for other studies, the study should mainly rely on up-to-date datasets, such as ERA5 (already used, partially). Please move the legacy reanalysis contents to the supplement, or remove; any difference should anyway be explained. The period of analysis needs to be given, currently many dates are provided for the various products, which is confusing. The authors mention using two PM2.5 products, but the results seem to agree only qualitatively. How do the products differ?

**Response:** Thank you very much for your valuable comments. In our earlier studies, we primarily used NCEP data, and to maintain consistency, we also employed NCEP in the initial version of the manuscript. However, we have now redrawn all the figures using ERA5 data, and these updated figures have been included in the main text (Figs. 4–10). All the original NCEP-based figures have been moved to the supplementary material (Figs. S5–S11). The results derived from ERA5 are largely consistent with those based on NCEP. We have clarified this point in the revised manuscript. In addition, we have revised the data time range in the manuscript according to your specific suggestion.

We used two PM2.5 datasets to verify the robustness of our results. As shown in Figures 2 and S3, the two datasets exhibit broadly consistent results, further demonstrating that the nonlinear modulation of the ENSO–PM2.5 relationship by the QBO identified in our study is robust. Notably, the two PM2.5 datasets were developed by different research teams and differ in both spatial resolution and data construction methods. The dataset provided by Yang (2020) was reconstructed using a spatiotemporal random forest model based on atmospheric visibility observations and other auxiliary data, which has a horizontal resolution of 1°×1° covers the period from 1980 to 2019. The dataset provided by Zhong et al.

(2022a) was reconstructed based on an advanced machine learning model, LightGBM using conventional meteorological observations, emissions, and elevation, which has a resolution of  $0.25^{\circ} \times 0.25^{\circ}$  and covers the period from 1960 to 2020.

**Lines 77–78:** "Further details on the dataset provided by Zhong et al. (2022a) please refer to Zhong et al. (2022b)."

Lines 80–87: "The monthly atmospheric reanalysis products, including geopotential height, zonal and meridional winds, and relative humidity, were obtained from the ERA5, the latest generation of the European Centre for Medium-Range Weather Forecasts (ECMWF). These reanalysis products have a horizontal resolution of  $0.25^{\circ}\times0.25^{\circ}$  and a vertical resolution of 37 levels from 1000 hPa to 1 hPa, starting from 1940. In addition, the reanalysis data from the NCEP–DOE AMIP-II reanalysis (Kanamitsu et al., 2002), provided by the National Centers for Environmental Prediction (NCEP)/National Center for Atmospheric Research (NCAR), are used to further validate the robustness of the results. The NCEP/NCAR atmospheric reanalysis dataset features a horizontal resolution of 2.5°  $\times$  2.5°, 17 vertical levels from 1000 hPa to 10 hPa, and spans the period from 1979 to the present."

Lines 262–263: "The analysis based on NCEP data also supports our conclusion (Figs. S5–11)."

**Lines 291–292:** "ERA5 atmospheric reanalysis data is available at https://cds.climate.copernicus.eu/datasets/reanalysis-era5-pressure-levels-monthly-means?tab=download (last access: 25 June 2025)."

The methodology should be improved, as it is unclear why the authors selected the Nov-Dec season. In the introduction, Nov-Jan is also mentioned.

**Response:** Thank you for your valuable comment. We apologize for the inconsistency regarding the season. In our study, we focus on the November–December (ND) period because this is when the most significant PM2.5 anomalies and the strongest influence of large-scale atmospheric circulations dominated by ENSO over the NCP are observed, based on both climatological analysis and prior studies (i.e., Ma et al., 2022; Zhao et al., 2022). Additionally, the QBO primarily influences the East Asian climate during the ND season (Ma et al., 2021). The mention of November–January (NDJ) in the introduction was based on previous studies that found ENSO can influence air pollution over North China during this broader early winter period. However, the QBO's impact on the East Asian climate is

mainly observed in ND, as shown in prior research (Ma et al., 2021). Therefore, this study focuses specifically on the ND season to better capture the combined influence of ENSO and QBO. To avoid confusion, we have added further clarification in the revised manuscript.

**Lines 99–101:** "Given that both ENSO and QBO exert their strongest climatic impacts on East Asia during November and December (ND), this study particularly emphasizes the season of ND, as highlighted in previous studies (Ma et al., 2021; Zhao et al., 2022)."

Given that nonlinear diagnostics are computed, the authors need to explain if high-resolution data (e.g., at least daily) were used, as signals would otherwise be smoothed.

Response: Thank you for this valuable comment. We apologize for not clarifying this point earlier. As shown in Fig. R1, we now use high-resolution daily PM2.5 data to compute the nonlinear diagnostics, in order to better capture the day-to-day variability and avoid excessive temporal smoothing. Overall, during the QBO easterly phase years, daily PM2.5 concentrations over the NCP show a marked increase during El Niño events (+5.25 μg m-3) and a significant decrease during La Niña events (-4.47 μg m-3). In contrast, during the QBO westerly phase years, PM2.5 concentrations over the NCP also change during El Niño (-2.84 μg m-3) and La Niña events (+2.30 μg m-3), but with much smaller magnitudes (approximately about 50% of those observed during the QBO easterly phase). This result further supports the conclusions obtained from the monthly data analysis. However, we acknowledge that in the high-resolution (daily) data, the nonlinear modulation effect of the QBO appears somewhat weaker, which could be related to the presence of more high-frequency and local signals in daily data. We have now clarified this point in the revised manuscript. In addition, future studies could further explore these interesting issues in detail.

**Lines 140–141:** "These conclusions are also supported by high-resolution data, such as daily observations (Fig. S2)."

**Lines 282–284:** "Another limitation of the current study is the lack of high-resolution data to further investigate the nonlinear processes identified, which remains an important direction for future research."

Figure R1: (a) Composite daily PM2.5 time series (μg m-3) during November–December for La Niña & EQBO (blue solid line), La Niña & WQBO (blue dashed line), El Niño & EQBO (red solid line), El Niño & WQBO (red dashed line). (b, c) Same as Figure 3 in the main text, but based on daily PM2.5 data.

The authors ascribe PM2.5 changes to changes in atmospheric circulation. However, my understanding is that haze conditions can be extremely local, e.g. due to the presence of urban settlements in valleys.

The discussion needs to be enhanced to explain relevant local features, including sources. Relevant analysis should be included, to understand the relative contributions of meteorology but also of primary and secondary sources. The motivation of section 4 is lacking. Why do these diagnostics matter?

Response: Thank you for your very valuable comment. Emission sources play a crucial role in the variation of PM2.5 concentrations (Dang and Liao, 2019). However, the interannual variability of PM2.5 concentration over eastern China is primarily influenced by meteorological conditions (Dang and Liao, 2019; Fig. R2). Therefore, our current study mainly focuses on investigating the impact of meteorological factors and their associated large-scale atmospheric circulations on PM2.5 concentrations. In this study, to reduce the potential impact of emissions on the interannual variability of air pollution, the least-squares linear trend was removed from the original PM2.5 dataset. We plan to further quantify the respective contributions of meteorological conditions and anthropogenic emissions to PM2.5 concentrations during different ENSO and QBO phases in future work. This is indeed a highly worthwhile topic to explore. Furthermore, given that local factors such as urban-scale effects exhibit relatively limited variability, these aspects were not the focus of the present study. Nonetheless, they warrant more detailed investigation in future research. Once again, we deeply appreciate your forward-looking comment. In addition, to clarify the motivation behind our focus on meteorological conditions, we have added some details in Section 4.

Figure R2: Time series of standardized PM2.5 concentration anomalies for November–December. The

red line represents the regionally averaged PM2.5 concentration anomaly over the NCP (32°–42°N, 112°–120°E) based on the PM2.5 dataset provided by Yang (2020). The blue line shows PM2.5 concentration anomalies due to emissions over China, derived from the Multi-resolution Emission Inventory model for Climate and air pollution research (MEIC), available at http://meicmodel.org.cn/?page id=541&lang=en. The climatology is based on the period 1990–2019.

**Lines 78–79:** "To minimize the influence of emissions on the interannual variability of air pollution, we removed the least-squares linear trend from the original PM2.5 data in this study."

**Lines 183–184:** "Given the important role of meteorological conditions and their associated large-scale circulations in air pollution, this section mainly elaborates on the influence of meteorological factors (Dang and Liao, 2019; Yin et al., 2021)."

Lines 281–282: "... GEOS-Chem are not included in the current manuscript, which can capable of quantifying the respective contributions of emissions and meteorology."

The authors need to clarify how their work is distinct from previous papers (several papers from the first author are cited) and what is the new understanding, see e.g. the non-exhaustive list of works below (none cited, as far as I can see).

Based on a simple online search, a number of references seem missing:

An et al. 2018 https://acp.copernicus.org/articles/18/1863/2018/ on ENSO and winter pollution in China

Jiang et al. 2017 https://link.springer.com/article/10.1007/s13351-017-6412-z on drivers on haze in China

Li et al. 2023 https://doi.org/10.5194/acp-23-1533-2023 on summer ozone pollution in China and QBO Lu et al. 2022 https://doi.org/10.1029/2022JD036938 on ENSO and EASM and summer ozone pollution in China

Ray et al. 2020 https://www.nature.com/articles/s41561-019-0507-3 on QBO and trace gases Sun et al. 2018 https://doi.org/10.1029/2018JD028534 on ENSO and winter pollution in East China Zhang et al. 2025 https://doi.org/10.1029/2024JD041825 for a model study of QBO/ENSO influence on ozone transport

Zhao et al. 2016 https://www.nature.com/articles/srep27424 on decadal variability of haze in China

**and tropical Pacific**

**Response:** We sincerely thank the reviewer for this valuable comment. We apologize for the oversight in not properly citing relevant previous works, including several studies closely related to our topic. In the revised manuscript, we have added citations to the missing references identified by the reviewer, as well as other relevant literature found through further review. We have also clarified how our study differs from prior work, including those by the first author, and emphasized the new understanding and contributions of this study. These clarifications are presented in the Introduction and Discussion sections, and all additions have been highlighted in the revised manuscript.

For example, compared with the study by Zhao et al. (2018), which suggested little to no influence of ENSO on wintertime air quality over the North China Plain, our work highlights a significant ENSO impact—particularly during early winter (November–December, ND). This apparent discrepancy may stem from differences in the analyzed time periods. While Zhao et al. focused on the full winter season (December–February, DJF), our study isolates the pre-winter period, during which the ENSO signal exerts a more pronounced influence on regional air quality. Liang and Tao (2017) emphasized the potential role of the QBO westerly phase in autumn in modulating haze days over Beijing, Tianjin, and Hebei, primarily through its influence on thermodynamic conditions. In contrast, our study focuses on the QBO easterly phase and its nonlinear modulation of the ENSO-PM2.5 relationship, offering a distinct perspective on the underlying mechanisms. Ray et al. (2020) reported the influence of the QBO on global surface trace gases, but their analysis did not focus on specific regions or address localized air quality issues. Similarly, Sun et al. (2018) primarily examined the individual effect of ENSO on aerosol variability over North China. Our work builds on this by emphasizing how the OBO modulates ENSO's influence on PM2.5 concentrations over the North China Plain during early winter, thereby revealing important nonlinear interactions between these large-scale climate modes. Moreover, Zhao et al. also focused on the decadal variability of haze days, whereas our analysis targets interannual variability in PM2.5 concentrations, highlighting different temporal scales of variability. Our previous studies primarily examined the roles of ENSO and Arctic sea-ice in affecting air quality. The present work advances this line of research by incorporating the QBO as a critical factor, adding a new dimension to our understanding of climate-pollution interactions. Finally, studies by Lu et al. (2022), Li et al. (2023), and Zhang et al. (2025) focused mainly on the drivers of summertime ozone variability, which is fundamentally different from our focus on wintertime PM2.5 pollution. Therefore, these works were not discussed in detail. These distinctions have now been clearly stated and incorporated into the revised manuscript.

Line 41: "... ENSO and air pollution over the NCP (e.g., Chang et al., 2016; Jeong et al., 2018; Sun et al., 2018; ..."

Lines 43–44: "... while others argued there is little to no correlation (e.g., Li et al., 2017; Zhao et al., 2018a, 2018b; Cheng et al., 2019; He et al., 2019)."

Lines 54–55: "... and regional air quality (Lian and Tao, 2017; Lu et al., 2022; Li et al., 2023; Zhang et al., 2025)."

**Lines 270–272:** "Compared with our earlier studies (2022a, 2022b, 2023a), this paper further emphasizes the important role of the QBO in influencing the key atmospheric circulation systems responsible for variations of PM2.5 concentrations over the NCP."

**Lines 284–289:** "In addition, Ray et al. (2020) pointed out that the QBO influences global surface trace gases. Therefore, it would be worthwhile for future studies to investigate how PM2.5 concentrations in regions beyond the NCP—such as India—are affected by the combined effects of ENSO and the QBO. Moreover, Zhao et al. (2016) found that the Pacific Decadal Oscillation (PDO) can influence the number of haze days in eastern China at a decadal time-scale. Therefore, whether the PDO modulates the impacts of ENSO and the QBO on PM2.5 concentrations over the NCP in early winter is a topic worthy of further investigation."

The authors are advised to revised the text with care to correct various typos and unsupported statements, a few reported in the list below. Also the text structure could be improved to avoid long paragraphs and focus the discussion.

**Response:** We sincerely thank the reviewer for the helpful comments and suggestions. In response, we have thoroughly revised the manuscript to correct the typographical errors and remove or better support previously unsupported statements. We also improved the text structure by breaking down overly long paragraphs and focusing the discussion to enhance clarity and readability. All changes have been highlighted in the revised manuscript.

1. L32 This reference is not suitable. Are there peer-reviewed papers on this 'comprehensiveness'? E.g. comparing in-situ or satellite-based emissions/concentrations?

**Response:** Thank you for your insightful comment. We acknowledge that the original reference may not be the most appropriate to support the claim regarding comprehensiveness. In response, we have replaced it with a peer-reviewed study that utilizes in-situ measurements.

**Lines 30–31:** "This has persisted despite the Chinese government's comprehensive emission control measures implemented since 2013 (Silver et al., 2025)."

2. L40 unclear 'its the crucial'

**Response:** Thank you for your careful review. We are sorry for this error. It should be "its crucial role in". We have changed it in the revised manuscript.

Line 39: "Given its crucial role in shaping global weather and climate, ..."

3. L44 I don't get this, which would be a review?

**Response:** Thank you for your comment. We apologize for the confusion. What we mean is that recent studies, building on previous conflicting viewpoints, have found that ENSO does indeed affect air pollution over the NCP. To prevent any misunderstanding, we have amended the original phrasing.

Line 43: "Recently, building on a synthesis of these conflicting findings, ..."

4. L47 You cannot expect the reader to know in detail the regional geography. Please include a maps with all the relevant names and features mentioned, perhaps in the supplement.

**Response:** Thank you for your helpful suggestion. We agree that readers may not be familiar with the detailed regional geography mentioned in the text. To address this, we have added a supplementary map showing all relevant geographical names and features (e.g., provinces, major cities) mentioned in the supplement. This addition will help improve clarity and allow readers to better understand the spatial context of our study.

Line 46: "... ENSO impacts air pollution in the Beijing-Tianjin-Hebei region (Fig. S1) ..."

Figure S1: Map for Hebei, Beijing and Tianjin of China.

**5. L64 typo 'underling'**

**Response:** Thank you for catching this typo. We have corrected "underlying" to "underlying" in Line 64 of the revised manuscript.

Line 65: "... composites and explore the underlying physical mechanisms."

6. L70 there is virtually no description of the data. Is it in-situ? Satellite? Model? The zenodo repository does not show the information neither. The link should be given elsewhere, while here readable information.

**Response:** Thank you for pointing this out. We agree that the original manuscript lacked sufficient detail regarding the source and nature of the PM2.5 dataset. In the revised manuscript, we have now clarified that the dataset is a model-reconstructed gridded product, derived using a spatiotemporal random forest model trained on atmospheric visibility observations (in-situ measurements) and auxiliary variables (e.g., meteorological data, land use, emission, population). The dataset itself does not come from direct satellite observations or chemical transport models, but from a machine-learning-based reconstruction approach. Additionally, we have moved the Zenodo link to a more appropriate section (e.g., Data Availability) and provided a concise but readable description of the dataset in the Data section to improve clarity for readers.

Lines 71–76: "The monthly gridded near-surface PM2.5 dataset provided by Yang (2020) has a horizontal resolution of 1°×1° and covers the period from 1980 to 2019. This dataset was reconstructed by Li et al. (2021) using a spatiotemporal random forest model based on atmospheric visibility observations and other auxiliary data. According to Li et al. (2021), the monthly PM2.5 concentrations show excellent agreement with ground-based measurements, with a coefficient of determination of 0.95 and a mean relative error of 12%. This dataset has been widely used in studies of air pollution in China (e.g., An et al., 2022b, 2022c; Zhang et al., 2023, 2024)."

7. L80 The expression 'Fifth major global reanalysis' makes no sense. Is there a minor version as well? The information on the spatial resolution and temporal coverage are both incorrect, you probably refer to what you retrieved. Also the reference is to the website, not the paper.

**Response:** Thank you for your valuable comments. We agree that the expression "Fifth major global reanalysis" is unclear and potentially misleading. We have revised the text to refer to "ERA5, the latest generation of ECMWF reanalysis" to improve clarity and avoid confusion. Regarding the spatial resolution and temporal coverage, we appreciate the reviewer's observation. We have corrected the description in the manuscript to reflect the official specifications of the full ERA5 dataset, rather than just the subset we retrieved. The revised version now accurately states the native horizontal resolution

 $(0.25^{\circ} \times 0.25^{\circ})$  and temporal coverage (from 1940 onward). We have also added the website. We thank the reviewer again for pointing out these important issues, which have helped us improve the accuracy and clarity of the manuscript.

**Line 87–88:** "Monthly boundary layer height data were downloaded from the ERA5, with a horizontal resolution of  $0.25^{\circ} \times 0.25^{\circ}$  at a single pressure since 1940."

8. L86 10 hPa is quite high, and here you are looking at instantaneous signals. Why not using 50 or 70 hPa to characterize a tropospheric process? Moreover, periods 2015/16 and 2019/20 showcase distinct QBO evolutions, which should be acknowledged

Response: We appreciate the reviewer's insightful comment. We agree that 10 hPa is located in the upper stratosphere, which may not directly reflect tropospheric processes. A fixed reference level of 10 hPa was chosen as it is closest to the level where the QBO amplitude is a maximum (metric  $h_{\text{max}}$  in Schenzinger et al., 2017) and often used in previous studies to represent the QBO phase (i.e., Bushell et al., 2022). As shown in Fig. R3, the QBO at 10 hPa exhibits a more regular quasi-biennial oscillation pattern with stronger amplitude compared to at 50 hPa (Fig. R4), which allows for the inclusion of a larger number of cases in our analysis. Regarding the second point, we agree that the QBO evolution during 2015/16 and 2019/20 was anomalous. In the analysis of physical mechanisms in the current manuscript, the period 2019/2020 was not selected. Additionally, removing the period 2015/2016 would not affect our conclusions as shown in Fig R5c. We have now acknowledged and discussed these features in the revised manuscript.

**Lines 151–153:** "It is worth mentioning that the QBO during the periods 2015/2016 and 2019/2020 exhibited distinct evolutions. Removing the QBO data for these periods does not affect our main conclusions (not shown)."

Figure R3: Nov-Dec averaged QBO index at 10 hPa (a) and 50 hPa (b).

Figure R4: Profile of QBO in the easterly phase. Figure is from https://acdext.gsfc.nasa.gov/Data\_services/met/qbo/qbo.html#singau.

Figure R5: Composite patterns of anomalous PM2.5 concentrations (shading; unit: μg m-3) for (a) La Niña & EQBO, (b) La Niña & WQBO, (c) El Niño & EQBO, (d) El Niño & WQBO, (e) nENSO & EQBO, (f) nENSO & WQBO. (g) and (h) Regression coefficients of anomalous PM2.5 concentrations onto the Niño3 index (QBO signals removed) and Niño3 index (the original data) shown in Fig. 1b–c. White dots (black grids in (a)–(f)) indicate areas of significance at the 80% (90%) confidence level. The period 2015/16 was removed.

9. L87 Based on the plots, there is one point per year. Is this an average of Nov-Dec? This needs to be explained, and points added in Fig. 1 b/c

**Response:** Thank you for your valuable comment. Yes, each point represents the average value over Nov-Dec for each year. We have now clarified this in the revised manuscript (L86–L87). In addition,

we have updated Fig. 1b and Fig. 1c to explicitly mark the data points corresponding to each year, as suggested.

Lines 94–95: "The mean wind speeds for November–December greater than ..."

**Figure 1:** "... averaged in the early winter months (Nov–Dec), each point represents the Nov–Dec average for a given year; ..."

**Figure 1:** (a) Time-height cross-section of the monthly mean equatorial zonal winds provided by the FUB (unit: m s-1); (b–c) Time series of the 10-hPa QBO and Niño3 indices, averaged in the early winter

months (Nov-Dec), each point represents the Nov-Dec average for a given year; (d) Regression coefficients of SST anomalies (unit: °C) onto the Niño3 index shown in (c). White dots indicate where values are significant at the 99% confidence level.

10. L89 The methodology of "Jan Null" is not explained. Please use some method justified in the literature

**Response:** Thank you. According to Jan Null, the Oceanic Niño Index (ONI) is the de facto standard used by NOAA to classify El Niño (warm) and La Niña (cool) events in the eastern tropical Pacific. It is defined as the running three-month mean sea surface temperature (SST) anomaly in the Niño 3.4 region (5°N–5°S, 120°–170°W). An El Niño event is identified when the SST anomaly equals or exceeds +0.5°C for five consecutive overlapping three-month periods, while a La Niña event is defined by anomalies at or below –0.5°C over the same duration. Given the paper's length, we have added these details in the Supplement Information and added some description in the revised manuscript.

**Line 97–98:** "... the website provided by Jan Null (see https://ggweather.com/enso/oni.htm and Text S1 for more details)."

11. Table 1 caption: period should be lowercase

**Response:** Thank you for your careful checking. We are sorry for this error. It has been changed in the revised manuscript.

**Table 1:** "Years of QBO and ENSO events based on the QBO index and ENSO events in November to December during the period ..."

12. Fig.1 are the units of regression coefficient degC as reported?

**Response:** Yes, it is "degC". The unit has been added in the caption of Fig. 1 in the new text.

Figure 1: "... (d) Regression coefficients of SST anomalies (unit: °C) onto ..."

13. L104 The period used should be explained

**Response:** Thank you. The period is 1979–2020. We have added the period in the revised text.

Line 113: "... the removal of the climatological mean for the period 1979–2020."

**14. L117 typo 'clearly'**

**Response:** Thank you for your careful checking. It should be 'evident'. We have revised it.

Line 132: "It is evident that ..."

**15. Fig. 2 one among g or h should be for the QBO?**

**Response:** We sincerely apologize for the confusion. Figure 2g shows the regression of PM2.5 concentrations on Niño3 index after linearly removing the QBO signal, while Figure 2h presents the regression of PM2.5 concentrations on the original Niño3 index. We have added clarification on this point in the revised manuscript.

**Figure 2:** "... (g) and (h) Regression coefficients of anomalous PM2.5 concentrations onto the Niño3 index (QBO signals removed) and Niño3 index (the original data) shown in Fig. 1b–c."

16. Fig. 3 I do not understand why you have "-1" subscripts. Are these referring to lagged values?

**Response:** We apologize for the confusion caused. This was an error in plotting, and we have corrected it in the revised manuscript, as shown below.

**Figure 3:** The average-mean PM2.5 concentration anomalies over the NCP (32°N–42°N, 110°E–120°E) for (a) La Niña & EQBO and La Niña & WQBO; (b) El Niño & EQBO and El Niño & WQBO. Boxplots are drawn based on 1,000 resamples using the bootstrap method.

**17. L142 You are presumably referring to Fig. 5**

**Response:** We appreciate your careful review and apologize for the error. It should indeed refer to Fig. 5, and we have corrected this in the revised manuscript.

Line 166: "... experiences higher (lower) boundary layer heights (Fig. 5a and 5c), ..."

18. Fig. 5 Units are not indicated anywhere

**Response:** We sincerely appreciate your thorough review and apologize for the omission of the unit. The units of boundary layer height anomalies (unit: m) and relative humidity anomalies (unit: %) have been added in the revised manuscript.

**Figure 5:** "Same as Figure 4, but for (a–d) boundary layer height anomalies (unit: m) and (e–f) 925-hPa relative humidity anomalies (unit: %)."

19. L165 by whom?

**Response:** Thank you for your careful checking. It is conducted by An et al. (2022b). We have added more details in the revised manuscript.

Line 193: "... anomaly (NAAA) by An et al. (2022b)) ..."

20. Fig. 6 This figure definitely differs from Fig. 4. Verify

**Response:** Thank you for your careful review. We have revised the caption of Fig.6 in the revised manuscript.

**Figure 6:** "Composite patterns of geopotential height anomaly at 500 hPa (shading and contours; unit: m) for (a) La Niña and EQBO, (b) La Niña and WQBO, (c) El Niño and EQBO, and (d) El Niño and WQBO."

21. Fig. 7 'disturbed' in what sense?

**Response:** We apologize for the confusion. It refers to the perturbation streamfunction, and we have revised the expression accordingly.

Line 521: "... but for the perturbation streamfunction ..."

22. L187 A similar statement was already done before

**Response:** Thank you for your careful checking. We have removed this sentence in Line 215.

23. Fig. 8 is hard to read and needs to be improved graphically

**Response:** Thank you. We have modified Figure 6 by reducing the contours density and intensifying

the color fill, as shown below.

**Figure 8:** Same as Figure 6, but for climatological zonal winds (green contours; unit: m s-1; interval: 5) and zonal wind anomalies (shading; unit: m s-1) at 300 hPa. White dotted (black grid) areas indicate significant values at the 80% (90%) confidence level. Purple boxes represent the key regions of zonal wind changes.

**24. L203 what's OBO now?**

**Response:** Thank you. We apologize for this error. It should be "QBO". We have revise it.

Line 229: "... between the QBO and extratropical circulations ..."

**25. L211 Unclear sentence**

**Response:** Thank you for your careful review. We have revised this sentence.

**Lines 237–238:** "These anomalous circulations modulate local meteorological conditions, thereby affecting PM2.5 concentrations over the NCP."

26. L226 The 'Eurasia hinterland' sounds odd

**Response:** Thank you. We have changed it.

Line 252: "... over subtropical Eurasia, ..."

27. L237 All this discussion does not seem relevant.

**Response:** Thank you. The equation in Fig. S2 is primarily derived from the work of An et al. (2023a), with the key modification being the inclusion of QBO as an additional explanatory variable. Accordingly, we discuss their study here to highlight the progress made in the present work relative to previous research.

28. L252 What do you mean? What numerical model? A transport model?

**Response:** We sincerely apologize for any confusion caused. The numerical model referenced here may refer to either WRF-Chem or GEOS-Chem, both of which are encouraged for use in future research investigating the relationship between QBO and PM2.5. We have added more details in the revised manuscript.

Lines 280–281: "... the validation using numerical models such as WRF-Chem or GEOS-Chem are not included ..."

29. L260 the link for the QBO has moved, verify. Links have been visited one year ago or more!

**Response:** We appreciate your careful review. Upon verification, we found that the link for the QBO remains valid (https://www.geo.fu-berlin.de/en/met/ag/strat/produkte/qbo/index.html).

**Line 298:** "... https://www.geo.fu-berlin.de/en/met/ag/strat/produkte/qbo/index.html (last access: 11 June 2025)."

30. L263 I can't locate experimental results in this work

**Response:** Thank you for your careful review. We are sorry for this confusion. We have revised this description.

Line 302: "XA, LS and WC designed the research and carried them out."

31. Fig. S2 Since all terms have an 'I', it could be safely omitted. What is the 'forecast data' mentioned? Is it again ERA5?

**Response:** Thank you for your careful checking. We have removed "I" as you suggested. In addition, the 'forecast data' is geopotential height anomaly of seasonal forecast anomalies on pressure levels from ERA5. We have added more details in the revised manuscript.

Fig. S12: "... based on forecast data of ERA5 provided by ..."

**Figure S12:** Linear regression of the time-series of the first EOF (represent the northeast Asian anomalous anticyclone (NAAA)) of November to January mean geopotential height anomalies at 500 hPa over the domain 25°–55°N, 100°–160°E during 1979–2019, with multiple variables including the Barents–Kala Sea sea-ice index (BKSI), Niño3 index (Niño3I) and 30-hPa QBO index (QBO30hPaI). The red solid line represents the NAAA index (500-hPa geopotential height anomaly averaged over 25°N–50°N, 120°E–160°E) based on reanalysis data, the blue solid line is the re-NAAAI reconstructed by multiple linear regression, the black dashed line is BKSI, the black dotted line is Niño3I, and the black dash-dotted line is QBO30hPaI. The black thick line represents 500-hPa geopotential height anomaly averaged over 25°N–50°N, 120°E–160°E based on forecast data of ERA5 provided by the

**References**

- Bushell, A.C., Anstey, J.A., Butchart, N., Kawatani, Y., Osprey, S.M., Richter, J.H., Serva, F., Braesicke, P., Cagnazzo, C., Chen, C.-.-C., Chun, H.-.-Y., Garcia, R.R., Gray, L.J., Hamilton, K., Kerzenmacher, T., Kim, Y.-.-H., Lott, F., McLandress, C., Naoe, H., Scinocca, J., Smith, A.K., Stockdale, T.N., Versick, S., Watanabe, S., Yoshida, K. and Yukimoto, S.: Evaluation of the Quasi-Biennial Oscillation in global climate models for the SPARC QBO-initiative, QJR Meteorol Soc, 148, 1459–1489, doi:10.1002/qi.3765, 2022.
- Li, M., Yang, Y., Wang, H., Li, H., Wang, P., and Liao, H.: Summertime ozone pollution in China affected by stratospheric quasi-biennial oscillation, Atmos. Chem. Phys., 23, 1533–1544, doi:10.5194/acp-23-1533-2023, 2023.
- Liang, J., and Tang, Y.: Climatology of the meteorological factors associated with haze events over northern China and their potential response to the Quasi-Biannual Oscillation, J. Meteorol. Res., 31, 852–864, doi:10.1007/s13351-017-6412-z, 2017
- Lu, S., Gong, S., Chen, J., He, J., Lin, Y., Li, X., et al.: Composite effects of ENSO and EASM on summer ozone pollution in two regions of China, Journal of Geophysical Research: Atmospheres, 127, e2022JD036938, doi:10.1029/2022JD036938, 2022.
- Ma, T., Chen, W., Huangfu, J., Song, L., and Cai, Q.: The observed influence of the Quasi-Biennial Oscillation in the lower equatorial stratosphere on the East Asian winter monsoon during early boreal winter, Int. J. Climatol., 41(14), 6254–6269, doi:10.1002/joc.7192, 2021.
- Ma, T., Chen, W., Chen, S., Garfinkel, C., Ding, S., Song, L., Li, Z., Tang, Y., Huangfu, J., Gong, H. and Zhao, W.: Different ENSO Teleconnections over East Asia in Early and Late Winter: Role of Precipitation Anomalies in the Tropical Indian Ocean and Far Western Pacific, Journal of Climate 35(24), 7919–7935, doi:10.1175/JCLI-D-21-0805.1, 2022.
- Ma, T. J., Chen, W., An, X. D., Garfinkel, C. I., and Cai, Q. Y.: Nonlinear effects of the stratospheric Quasi-Biennial Oscillation and ENSO on the North Atlantic winter atmospheric circulation, J. Geophys. Res. Atmos., 128, e2023JD039537, doi:10.1029/2023JD039537, 2023.

- Zhao, W., Chen, S., Zhang, H., Wang, J., Chen, W., Wu, R., Xing, W., Wang, Z., Hu, P., Piao, J., and Ma, T.: Distinct Impacts of ENSO on Haze Pollution in the Beijing–Tianjin–Hebei Region between Early and Late Winters, J. Climate, 35, 687–704, doi:10.1175/JCLI-D-21-0459.1, 2022.
- Ray, E.A., Portmann, R.W., Yu, P. et al.: The influence of the stratospheric Quasi-Biennial Oscillation on trace gas levels at the Earth's surface, Nat. Geosci. 13, 22–27, doi:10.1038/s41561-019-0507-3, 2020.
- Schenzinger, V., Osprey, S., Gray, L.J. and Butchart, N.: Defining metrics of the Quasi-Biennial Oscillation in global climate models, Geoscientific Model Development, 10, 2157–2168, 2017.
- Sun, J., Li, H., Zhang, W., Li, T., Zhao, W., Zuo, Z., et al.: Modulation of the ENSO on winter aerosol pollution in the eastern region of China, Journal of Geophysical Research: Atmospheres, 123, 11,952–11,969, doi:10.1029/2018JD028534, 2018.
- Zhang, Z., Wang, Z., Liang, J., and Luo, J.: Impacts of the Quasi-Biennial Oscillation and the El Niño-Southern Oscillation on stratosphere-to-troposphere ozone transport: Assessment with chemistry-climate models, Journal of Geophysical Research: Atmospheres, 130, e2024JD041825, doi:10.1029/2024JD041825, 2025
- Zhao, S., Li, J., and Sun, C.: Decadal variability in the occurrence of wintertime haze in central eastern China tied to the Pacific Decadal Oscillation, Sci. Rep., 6, 27424, doi:10.1038/srep27424, 2016.
- Zhao, S., Zhang, H., and Xie, B.: The effects of El Niño–Southern Oscillation on the winter haze pollution of China, Atmos. Chem. Phys., 18, 1863–1877, doi:10.5194/acp-18-1863-2018, 2018.

---

## Author Response (AR2)

Dear reviewer #1,

We sincerely appreciate your careful review of our manuscript and your valuable suggestions for improving the paper. We have thoroughly considered all comments and revised the manuscript accordingly. Below are our point-by-point responses. *Italicized text* indicates the reviewers' comments, while the regular text represents our responses. The specific revisions are highlighted in red, and all corresponding changes have been marked in the manuscript in the same manner.

Sincerely,

Xiadong An

On behalf of all authors

**Anonymous Referee #1**

In the revised manuscript "Nonlinear effects of the stratospheric Quasi-Biennial Oscillation on ENSO modulating PM2.5 over the North China Plain in early winter" by An et al. my comments were adequately addressed.

The paper is therefore recommended for publication in ACP after addressing a few remaining technical comments.

**Response:** Thank you for your positive evaluation of our work and for recommending it for publication in ACP. We will carefully address all the remaining technical comments to further improve the manuscript.

**TECHNICAL COMMENTS:**

1. l.13: concentrations -> concentration

**Response:** Thank you for pointing this out. We have changed "is" to "are" and retained the use of "concentrations" in line 13.

2. l.58: winter climate -> and winter climate

**Response:** Thank you for the suggestion. We have revised "winter climate" to "and winter climate" in line 59.

3. l.77: Further -> For further

**Response:** Thank you for the comment. We have changed "Further" to "For further" in line 78.

4. l.135, 136: what do the numbers in parentheses mean? are these 1-sigma uncertainties?

**Response:** Thank you for your question. The numbers in parentheses indicate the PM2.5 concentration changes during the WQBO phase, corresponding to "WQBO" in the parentheses in the sentence.

*5. l. 281*: ???

which can capable of quantifying -> with the help of these models we would be capable to quantify

Response: Thank you for pointing this out. We have revised the sentence to "with the help of these

Supplement:

1. l.61: represent -> representing

models we would be capable to quantify" in line 287.

**Response:** Thank you for the suggestion. We have changed "represent" to "representing" in line 61.

2. l.63: Kala -> Kara

**Response:** Thank you for pointing this out. We have corrected "Kala" to "Kara" in line 63.

Dear reviewer #2,

We again appreciate your careful review of our manuscript and the valuable suggestions for improving our work. We have thoroughly addressed all comments and revised the manuscript accordingly. Below are our point-by-point responses. *Italicized text* represents your comments, while the regular text contains our responses. The specific revisions are highlighted in blue, and all corresponding changes have been marked in the manuscript in the same way.

Sincerely,

Xiadong An

On behalf of all authors

**Anonymous Referee #2**

The authors performed several changes to their original manuscript, but the current version still needs a number of improvements for clarity.

**Response:** Thank you for your constructive feedback. We have carefully revised the manuscript to improve clarity and address the issues raised.

1. My comment on nonlinear diagnostics was probably unclear; I am referring to the reanalysis-based quantities (e.g. EP fluxes). I believe daily PM2.5 cannot be regarded as "high resolution" (please revise L140 accordingly). The meaning of "high resolution" at L282 is also unclear: what do you mean? High frequency PM2.5 measurements?

**Response:** Thank you very much for your clarification, and we apologize for our insufficient understanding of your point. In fact, EP fluxes based on monthly data are also widely used (e.g., Ma et al., 2021). In this study, we mainly use EP fluxes to indicate the propagation direction of planetary waves, rather than to examine the propagation of eddy momentum fluxes and heat fluxes, which to some extent reduces the requirement for high temporal resolution data. In addition, high-temporal-resolution meteorological data may contain many weather signals of different scales as well as noise.

For example, when selecting PM2.5 pollution events corresponding to QBO and ENSO events, we can only use monthly data, since the currently available QBO and ENSO indices are provided only at a monthly resolution. If we analyze daily data for these monthly-scale pollution events, we can only examine the entire month, which may include signals of various scales and thus be unfavorable for drawing clear conclusions. That said, we acknowledge that you have raised a very thought-provoking point, and we will continue to explore this line of thinking in our future research.

We agree that daily PM2.5 cannot be considered "high resolution" at line 140 in this context, and have removed that reference. We also have clarified that "high resolution" at line 282 refers to high-frequency (daily or hourly) PM2.5 measurements, which can better capture short-term variations.

Lines 147–148: "These conclusions are also supported by daily observations (Fig. S2)."

Lines 289: "... limitation of the current study is the lack of high-resolution data (e.g., daily or hourly data) to further investigate the ..."

**2. My comment on current Fig. 9 (previous Fig. 8) was not addressed.**

**Response:** I apologize for the misunderstanding. In the previous revised version of the manuscript, we had made modifications to the Figure 8 in the current version but inadvertently overlooked Figure 9. In the latest version, we have also redrawn Figure 9. In addition, Figure S10 is also changed.

**Figure 9:** Climatological mean of the zonal-mean zonal winds (red contours with an interval of 10 m s-1) and composite zonal-mean zonal wind anomalies (blue contours with an interval of 2 m s-1) in 0°–360°E for (a) La Niña and EQBO, (b) La Niña and WQBO, (c) El Niño and EQBO, and (d) El Niño and WQBO. (e)–(f) Same as (a)–(d), but for zonal winds averaged in 0°–140°E. Solid (dashed) lines represent positive and negative values, respectively. Grey shaded areas indicate significant values of the composite zonal winds at the 90% confidence level.

3. Given the use of ERA5 across several places, the information on data availability needs to be revised. **Response:** Thank you for the comment. We have revised the data availability information to clarify the sources and access details. Specifically, ERA5 atmospheric reanalysis data at pressure levels are

available at https://doi.org/10.24381/cds.6860a573 (Hersbach et al., 2018; last accessed 25 June 2025). Monthly boundary layer height data, as a surface variable, were also obtained from ERA5 (https://doi.org/10.24381/cds.f17050d7; Hersbach et al., 2018; last accessed 6 September 2024). These revisions ensure consistency in reporting ERA5 usage throughout the manuscript.

**Lines 297–299:** "ERA5 atmospheric reanalysis data at pressure levels are available at https://doi.org/10.24381/cds.6860a573 (Hersbach et al., 2018; last access: 25 June 2025). Monthly boundary layer height data, as a surface variable, were also obtained from ERA5 (https://doi.org/10.24381/cds.f17050d7, Hersbach et al., 2018; last access: 6 September 2024)."

**Specific comments**

1. L31 The Silver's work gives a different message than yours, please revise

**Response:** Thank you for your comment. We have revised the text to more accurately reflect Silver's work.

**Lines 30–31:** "The declining trend of PM2.5 concentrations in recent years appears to be slowing, despite the Chinese government's comprehensive emission control measures implemented since the 2010s (Silver et al., 2025)."

2. L77 It seems that a "For" is missing, but I am not sure what you mean. At least the period of the dataset should be given.

**Response:** Thank you for your comment. We have added "For" at the beginning of the sentence and specified the period of the dataset to improve clarity.

Line 78: "For further ..."

Line 78: "... monthly PM2.5 data spanning 1960 to 2020, as provided by Zhong et al. (2022a, 2022b)."

3. L78 I don't see how detrending removes the influence of emissions; or do you mean doing a regression?

**Response:** Considering that China's emissions first increased and then decreased (as shown by the blue solid line in Figure RR1), to minimize the influence of this emission trend, we removed the quadratic

trend from the original data rather than simply using regression analysis to remove a linear trend. After removing the quadratic trend, the correlation coefficient between the observed  $PM_{2.5}$  concentrations and the  $PM_{2.5}$  concentrations from emissions is 0.08 (p-value = 0.69), whereas for the original observed  $PM_{2.5}$  concentrations, the correlation coefficient with emissions is 0.40 (p-value = 0.03). This indicates that removing the quadratic trend partially eliminates the influence of emissions on the observed  $PM_{2.5}$  concentrations. To avoid causing confusion for readers, we have added more details in the revised manuscript.

**Lines 81–84:** "After removing the quadratic trend, the correlation coefficient between the observed  $PM_{2.5}$  concentrations and the  $PM_{2.5}$  concentrations from emissions is 0.08 (p-value = 0.69), whereas for the original observed  $PM_{2.5}$  concentrations, the correlation coefficient with emissions is 0.40 (p-value = 0.03). This indicates that removing the quadratic trend partially eliminates the influence of emissions on the observed  $PM_{2.5}$  concentrations."

**Figure RR1:** Time series of standardized PM2.5 concentration anomalies for November–December. The red line represents the regionally averaged PM2.5 concentration anomaly over the NCP (32°–42°N, 112°–120°E) based on the PM2.5 dataset provided by Yang (2020). The red solid line represents the original values, while the dashed line represents the values after removing the quadratic trend. The blue line shows PM2.5 concentration anomalies due to emissions over China, derived from the Multi-resolution

Emission Inventory model for Climate and air pollution research (MEIC), available at http://meicmodel.org.cn/?page id=541&lang=en. The climatology is based on the period 1990–2019.

**4. L80 still missing a reference**

**Response:** Thank you for pointing this out. We have added the appropriate reference at line 87 in the revised manuscript to address this issue.

Line 87: "... (Hersbach et al., 2018)."

**5. L88 BLH is a surface variable, hence no pressure level**

**Response:** Thank you for the comment. We have removed the description regarding the pressure level at line 93.

**Lines 92–93:** "Monthly boundary layer height data were downloaded from the ERA5, with a horizontal resolution of  $0.25^{\circ} \times 0.25^{\circ}$  since 1940."

**6. L124 This paragraph does not seem motivated, provide context**

**Response:** Thank you for your comment. We agree that the motivation for this paragraph was not clearly stated. We have revised the paragraph to provide proper context by explaining why this analysis is conducted and how it relates to the main research question.

**Line 129:** "Given that PM2.5 concentrations over the NCP are affected by multiple factors, such as ENSO and the QBO, ..."

7. L127 "NAAA" should be a subscript of "re". I do not understand what are both terms. I understand "re-NAAA" has units of concentration, as PM2.5? Please explain how these coefficients are identified.

**Response:** Thank you for your comment. We apologize for the unclear notation. Here, "NAAA" refers to the northeast Asian anomalous anticyclone, which was identified by An et al. (2022, 2023). It plays a crucial role in influencing PM2.5 concentrations over the NCP and is considered an indicator of the pollution potential over this region. In this study, the positive geopotential height anomalies over northeast Asia shown in Figure 6c resemble the main pattern of the NAAA. Other factors in Eq. 5, such

as ENSO and Arctic sea ice, also reported by An et al. (2023), are important climate factors affecting the NAAA. In this study, we additionally include the QBO as a contributing factor. The prefix "re" stands for "reconstructed", meaning that we reconstructed the NAAA index using several climate factors including ENSO, QBO, and the Arctic sea ice index, instead of directly using the NAAA index obtained from EOF analysis as in An et al. (2022, 2023). The NAAA index represents atmospheric circulation and is expressed using the 500 hPa geopotential height, hence its unit is meters (m). Therefore, "re-NAAA" reflects the reconstructed NAAA, a key circulation for winter air pollution in the NCP. We have clarified the notation and added these details in the revised manuscript.

**Lines 133–134:** "Here, re-NAAA represents the reconstructed northeast Asian anomalous anticyclone index (unit: m), a key circulation pattern influencing PM2.5 pollution in the NCP, which was identified by An et al (2023a)."

**Lines 135–136:** "Among them, ENSO and Arctic sea ice have already been identified by An et al. (2023a) as key factors influencing the NAAA."

8. L280 I don't think models can be used for validation, maybe for further analysis?

**Response:** Thank you for your comment. We agree that numerical models may be more suitable for further analysis rather than strict validation. In our study, the main objective was to reveal the nonlinear role of the QBO in modulating the ENSO–PM2.5 relationship based on observations. Using numerical models such as WRF-Chem or GEOS-Chem and further analysis in future work could help to further investigate the underlying mechanisms and quantify the contributions of emissions and meteorological factors. We have revised it.

Line 287: "... and further analysis ..."

9. L300 "research" is singular, and "it out"

**Response:** Thank you for pointing this out. We have corrected the sentence so that "research" is treated as singular and revised "them out" to "it" for grammatical accuracy.

Line 308: "... carried it out."

10. L303 who is XD?

**Response:** Thank you for your comment. "XD" refers to XA, and we have revised the text to spell out the full name to avoid ambiguity.

Line 309: "XA prepared ..."

11. Text S1 I still can't understand why you can just refer to NOAA

**Response:** Thank you for your comment. The table of ENSO events used in this study was primarily compiled by Jan Null, based on NOAA's sea surface temperature data. This is likely because NOAA's dataset is among the most widely used sources for sea surface temperature. Therefore, we mainly refer to NOAA.

12. Fig.S1 caption is not grammatically correct

**Response:** Thank you for pointing this out. We have revised the caption to read: "Figure S1: Map showing Hebei, Beijing, and Tianjin, China."

13. Fig.S2 I do not understand if, as stated, this uses a different dataset from the paper

**Response:** Thank you for your comment. Figure S2 uses the same PM2.5 dataset (Yang, 2020) as the main text. It is included to provide additional visualizations and support the robustness of our results. We have clarified this point in the revised caption and supplementary text.

**Figure S2:** (a) Composite daily PM2.5 time series (μg m-3) during November–December for La Niña & EQBO (blue solid line), La Niña & WQBO (blue dashed line), El Niño & EQBO (red solid line), El Niño & WQBO (red dashed line). (b, c) Same as Figure 3 in the main text, but based on daily PM2.5 data. The daily PM2.5 data were provided by Yang (2020).

14. Fig.S6 I have the impression some plots for the main are repeated. This is confusing; results for the bottom plots are quite different from those in the paper, so these cannot be dismissed as currently done.

**Response:** Thank you very much for carefully reviewing our manuscript. Indeed, panels a–d in Figure S6 are consistent with the results shown in Figure 5a–d of the main text. In the original manuscript, we

used ERA5 boundary layer height data, while the other data were from NCEP. Following your suggestion to use ERA5 for all analyses, we have replaced the NCEP-based plots with ERA5-based ones in the previous revised version of the manuscript. The boundary layer height plots were not replaced, as they were already based on ERA5. To avoid causing any confusion, we have removed the boundary layer height panels from Figure S6.

**Figure S6:** Same as Figure 5e-h in the manuscript, but for NCEP data.

**References**

An, X., Chen, W., Hu, P., Chen, S., and Sheng, L.: Intraseasonal variation of the northeast Asian anomalous anticyclone and its impacts on PM2.5 pollution in the North China Plain in early winter, Atmos. Chem. Phys., 22, 6507–6521, https://doi.org/10.5194/acp-22-6507-2022, 2022.

An, X. D., Chen, W., Sheng, L. F., Li, C., and Ma, T. J.: Synergistic Effect of El Niño and Arctic Sea-Ice Increment on Wintertime Northeast Asian Anomalous Anticyclone and Its Corresponding PM2.5 Pollution, J. Geophys. Res. Atmos., 128, e2022JD037840, doi:10.1029/2022JD037840, 2023.

Ma, T., Chen, W., Huangfu, J., Song, L., and Cai, Q.: The observed influence of the Quasi-Biennial Oscillation in the lower equatorial stratosphere on the East Asian winter monsoon during early boreal winter, Int. J. Climatol., 41(14), 6254–6269, doi:10.1002/joc.7192, 2021.

---

## Author Response (AR3)

Dear Editor and Reviewer #2,

We sincerely thank you and Reviewer #2 for once again taking the time to carefully read our manuscript and previous responses, and for providing constructive feedback that has helped us to further improve our work. With the reviewer's thoughtful and detailed suggestions, the overall quality of the paper has been substantially enhanced. We also wish to express our gratitude to the editor for the careful consideration and recognition of our manuscript. We have carefully addressed all comments and revised the manuscript accordingly. Below, we provide our point-by-point responses. *Italicized text represents the reviewer's comments*, while the regular text contains our responses. All specific revisions are highlighted in blue and have been marked consistently throughout the revised manuscript.

Sincerely,

Xiadong An

On behalf of all authors

**Anonymous Referee #2**

I suggest more care when it comes to referencing and more transparency when communicating the limitations of the analysis.

**Response:** We sincerely thank you for your constructive suggestions. You are right, and we agree that we should be more careful in the two aspects you mentioned. In response to your specific comments, we have made corresponding revisions and provide detailed explanations below.

1. My point on EP fluxes computation should be acknowledged in the text. Citing only paper (by the authors) using monthly data as justification is unconvincing, and the effects of noise/weather would anyway be reduced if averaging after the calculation. This choice of starting from monthly data likely leads to a smoothing of resulting fields, and it could easily be solved by using daily or subdaily data (line 289). Perhaps only PM data is "lacking" at sub-monthly timescale. QBO and ENSO data can be computed also at daily scales, so this argument does not hold as well.

Response: Thank you for your valuable suggestion. We have recalculated the EP fluxes using daily ERA5 data, and the results are shown in Fig. R1. Our method was as follows: we first computed the EP fluxes from daily reanalysis data, then calculated their monthly means, and finally composited the EP flux distributions for different QBO–ENSO phase combinations (as defined in Table 1 of the main text) using monthly EP flux anomalies with the climatology removed (Fig. R1). The results indicate that the EP fluxes based on daily data are overall consistent with those derived from monthly data. For instance, the propagation direction of planetary waves remains the same. Minor differences appear mainly in the stratosphere, while in the tropospheric regions of our interest—such as the subtropical and subpolar areas (Fig. R1 a and c)—the EP flux divergences are also consistent. This further demonstrates that our results are robust. We have added corresponding details in the main text and included the figure in the Supplementary Materials.

Figure R1: Cross-sections of the EP flux (vectors; unit: m² s⁻²) based on daily ERA5 data for waves 1–2 and its divergence (contours, with red (blue) lines represent divergence (convergence); unit: m s⁻¹ day⁻¹) for (a) La Niña and EQBO, (b) La Niña and WQBO, (c) El Niño and EQBO, and (d) El Niño and WQBO. Heavy and light shaded areas indicate significant values at the 95% and 90% confidence levels, respectively.

**Lines 244–245:** "To further ensure the robustness of our conclusions, we recalculated the EP fluxes using daily reanalysis data and found the results to be largely consistent (Fig. S12)."

2. I don't see the need for mentioning "WRF-Chem or GEOS-Chem" at line 286, without reference. The reader might be confused by this. I guess you just mean using atmospheric models with interactive chemistry. Please revise for clarity.

**Response:** We thank the reviewer for this helpful comment. To avoid confusion, we have revised the sentence at line 286 accordingly and removed the unnecessary model names.

**Lines 287–288:** "... the validation using atmospheric models with interactive chemistry are not included in the current manuscript, ..."

3. The authors need to use the ERA5 paper citation in the text, i.e. Hersbach et al. 2020 (https://rmets.onlinelibrary.wiley.com/doi/10.1002/qj.3803) and not Hersbach et al. 2018, which is a website and not a paper.

**Response:** Thank you for your careful checking. We have made the necessary revisions.

Line 87: "... (ECMWF) (Hersbach et al., 2020)."

**Lines 371–372:** "Hersbach, H., Bell, B., Berrisford, P., et al.: The ERA5 global reanalysis, Q. J. R. Meteorol. Soc., 146, 1999–2049, doi:10.1002/qj.3803, 2020."

4. The quality of Fig. S1 is poor. Please use different colors, and explain in the caption if these are regions, cities or else. The reader might not know the geography of the area.

**Response:** We thank the reviewer for the suggestion. We have improved the quality of Fig. S1 by using more distinct colors and have clarified in the figure caption that the marked areas correspond to specific regions and cities. This should make the geographic information clearer for readers.

Figure S1: Map showing Hebei (yellow shading), Beijing (red shading), and Tianjin (blue shading), China. The area enclosed by the blue curve corresponds to the Beijing-Tianjin-Hebei region.

5. In line 69 of the Supplement, you mention "ERA5 forecast data". I guess you are using the analysis, not the forecast. Please confirm this and adjust here and elsewhere accordingly. In Fig. S12, the explanation for the grey shading is missing.

**Response:** We thank the reviewer for pointing this out. In line 69 of the Supplement, we indeed used the ERA5 seasonal forecast products (https://cds.climate.copernicus.eu/datasets/seasonal-postprocessed-pressure-levels?tab=download) provided by ECMWF, rather than the ERA5 analysis data. We have corrected the wording in the revised Supplement and carefully checked the manuscript to ensure consistency throughout. Regarding Fig. S12, we appreciate the reviewer's careful reading. We have now added the explanation that the grey shading indicates the period when hindcast experiments were conducted for verification.

Line 73 of the Supplement: "... based on ERA5 seasonal forecast products provided by ..."

Line 74 of the Supplement: "The gray shading indicates the period when hindcast experiments were carried out for verification."